# Instability and Local Minima in GAN Training with Kernel Discriminators

**Evan Becker**
Dept. CS
UCLA
evbecker@cs.ucla.edu

**Parthe Pandit**
HDSI
UC, San Diego
parthepandit@ucsd.edu

**Sundeep Rangan**
Dept. ECE
NYU
srangan@nyu.edu

**Alyson K. Fletcher**
Dept. Statistics
UCLA
akfletcher@ucla.edu

## Abstract

Generative Adversarial Networks (GANs) are a widely-used tool for generative modeling of complex data. Despite their empirical success, the training of GANs is not fully understood due to the min-max optimization of the generator and discriminator. This paper analyzes these joint dynamics when the true samples as well as the generated samples are discrete, finite sets, and the discriminator is kernel-based. A simple yet expressive framework for analyzing training called the *Isolated Points Model* is introduced. In the proposed model, the distance between true samples greatly exceeds the kernel width, so each generated point is influenced by at most one true point. Our model enables precise characterization of the conditions for convergence, both to good and bad minima. In particular, the analysis explains two common failure modes: (i) an approximate mode collapse and (ii) divergence. Numerical simulations are provided that predictably replicate these behaviors.

## 1 Introduction

Generative Adversarial Networks (GANs) are the most widely-used method for learning generative models of complex and structured data in an unsupervised manner. Indeed, GANs have seen incredible empirical success in a wide variety of domains ranging from image generation, speech generation, text generation, and many more. Models trained in this manner have also become critical in downstream applications. See [1, 29] for an overview.

In the GAN methodology, a *generator* model is trained to output samples that emulate a target dataset, which we call true samples. A critic model, called the *discriminator*, is trained to tell apart (discriminate) the true and generated samples. The generator is trained in parallel to fool the discriminator.

Correctly tuning the joint training of the discriminator and generator is one of the key challenges in GANs and is the source of several empirically observed problematic phenomena. For example, it is well-known that the resulting distributions can suffer from mode collapse and catastrophic forgetting. The optimization can also lead to divergence or slow convergence of min-max optimization algorithms. See [10, 19] for more details. Practical GANs methods overcome these issues with a combination of careful hyper-parameter optimization and heuristics. Significant effort has strived to develop theoretical frameworks that can better analyze and optimize GAN training.

36th Conference on Neural Information Processing Systems (NeurIPS 2022).

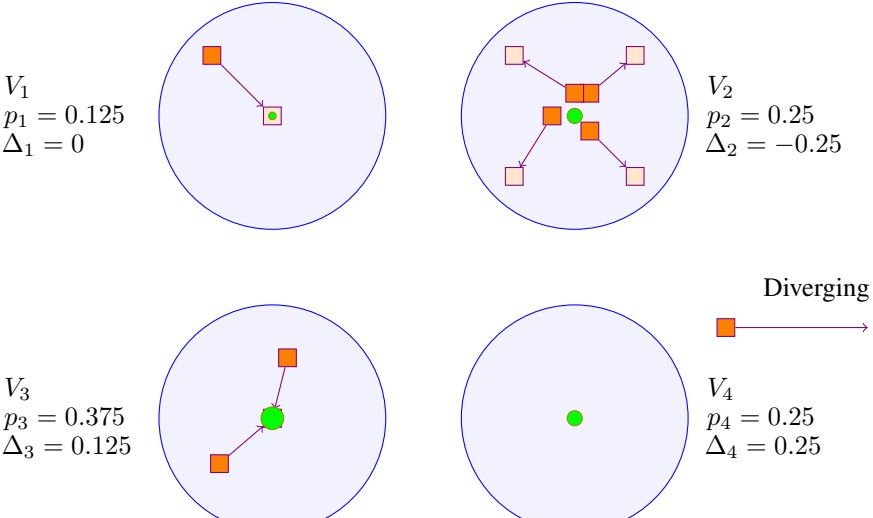

Figure 1: (**Isolated Points Model**) An illustration of our results. Each isolated region (blue circle) has a single true point (green disk) with point mass $p_i$. There are eight generated points with fixed point mass $\widetilde{p}_j = 1/8$ starting at the locations shown in the orange squares. In $V_1$ and $V_3$, the excess point mass is non-negative ($\Delta_i \geq 0$) and the generated points converge to the true point. In $V_2$, the excess point mass $\Delta_4 < 0$ and the generated points converge to a stable equilibrium around the true point analogous of mode collapse. Finally, a point outside all four isolated regions may diverge to $\infty$ in a linear velocity.

In this work, we propose a simple theoretical model, called the *Isolated Points Model*, that is analytically tractable and allows us to rigorously study the stability and convergence properties of training a GAN. In the proposed model, the true and generated points are discrete distributions over finite sets, and the discriminator is kernel-based meaning that it is linearly parametrized. We make an additional critical assumption that the true points are sufficiently separated such that the kernel interaction between points near two distinct true points is negligible. For distance-based kernels, this assumption essentially requires that the true points are separated much greater than the kernel width. A simple example of this model with four true points is illustrated in Fig. 1.

**Main Contributions** We show that this simple isolated points model provides sufficient richness to exhibit several interesting phenomena and insights:

(1) *Local stability and instability:* We provide necessary and sufficient conditions for local convergence of generated points to a true point within each isolated region (Theorem 1). The results show that the stability is determined by the excess point mass, meaning the difference between the true and generated mass in the region. A consequence of these results, Corollary 1, is that exact mode collapse where an excess of generated points concentrate in a single true point, is provably not possible.

(2) *Approximate mode collapse:* Although an excess of generated point mass cannot exactly concentrate on a single true point, we prove (see Theorem 2) the existence of locally stable equilibria where an arbitrarily large excess point mass concentrates *near* the true point. We call this phenomena *approximate mode collapse*.

(3) *Divergence:* We also demonstrate that given a initial perturbation, an isolated generated point can lock into an trajectory moving away from any true point in an arbitrary direction. Interestingly, this trajectory is driven solely by the generated point's own kernel history.

(4) *Role of the kernel width:* The results provide insights into the role of the kernel width in training – See Section 6. In particular, wider kernel widths reduce the likelihood of different modes of the distribution being isolated, which is the critical for the approximate mode collapse and divergence that we observe. At the same time, discriminators with wide kernels make are unable to distinguish points that are close, resulting in slow convergence near the true distribution.

**Prior work**  Convergence problems for GANs have been widely-recognized and studied since their inception [1, 29]. Indeed, many of the developments in GANs, notably the popular Wasserstein GAN and its variants, were motivated to overcome these issues [2, 3, 13, 17].

For analytic tractability, we focus on a relatively simple GAN with a kernel discriminator and maximum mean discrepancy loss. This methodology has been applied in other works such as [6, 9, 12, 27]. Our focus is on joint optimization of a kernel discriminator and a generator with multiple discrete points. An important avenue of future work would be to extend our results to more complex losses such as the Wasserstein loss [2, 3, 13].

The early Dirac-GAN [19], and its extension [26], considered a linear discriminator and a single Dirac-distribution for the generator and true data. Our model extends this work by considering multiple points and a general kernel discriminator. Importantly our model allows for a new parameter called the *excess point mass* which enables modeling complex behaviours in the isolated regions.

Our results also heavily rely on linearization methods derived from control theory, which have been studied in the GAN context in [21, 22, 30]. See, also general functional descriptions in [9, 16, 20, 23]. These prior works demonstrated the local stability of the generated distribution close to the true distribution. Stability results with a single-layer NN generator and linear discriminator has been studied in [4]. Due to the isolated points model, our results also prove the existence of locally stable *bad* local minima.

Metrics for understanding the performance of generative models remains an open challenge. Some efforts in this direction were presented in [8, 25], and we take advantage of this progress to report our numerical simulations.

## 2   Isolated Points Model

We propose the following model for studying the training of GANs.

**Discrete true and generated distributions:**  We assume that the true and generated distributions, $\mathbb{P}_r$ and $\mathbb{P}_g$, are over discrete sets in $\mathbb{R}^d$:

$$\mathbb{P}_r(\boldsymbol{x}) = \sum_{i=1}^{N_r} p_i \delta(\boldsymbol{x} - \boldsymbol{x}_i), \quad \mathbb{P}_g(\boldsymbol{x}) = \sum_{j=1}^{N_g} \widetilde{p}_j \delta(\boldsymbol{x} - \widetilde{\boldsymbol{x}}_j), \tag{1}$$

where $N_r$ and $N_g$ are the number of true and generated points, $\boldsymbol{X} = \{\boldsymbol{x}_i\}_{i=1}^{N_r}$ and $\widetilde{\boldsymbol{X}} = \{\widetilde{\boldsymbol{x}}_j\}_{j=1}^{N_g}$ are the true and generated points, respectively, and $\{p_i\}$ and $\{\widetilde{p}_j\}$ are their probabilities. For the generator, we assume that the probabilities $\widetilde{p}_j$ are fixed. The problem is to learn the point mass locations $\widetilde{\boldsymbol{X}}$ so that the generated and true distributions match.

**Kernel discriminator:**  We consider a GAN where the discriminator has a linear parametrization of the form:

$$f(\boldsymbol{x}, \boldsymbol{\theta}) = a(\boldsymbol{x})^\mathsf{T} \boldsymbol{\theta}, \tag{2}$$

for some vector of basis functions $a(\boldsymbol{x})$ and parameter vector $\boldsymbol{\theta}$. The discriminator is trained to maximize a maximum mean discrepancy (MMD) metric [6, 9, 12, 27]

$$\mathcal{L}_d(\boldsymbol{\theta}, \{\widetilde{\boldsymbol{x}}_j\}_{j=1}^{N_g}) := \sum_i p_i f(\boldsymbol{x}_i, \boldsymbol{\theta}) - \sum_j \widetilde{p}_j f(\widetilde{\boldsymbol{x}}_j, \boldsymbol{\theta}) - \frac{\lambda}{2} \|\boldsymbol{\theta}\|^2, \tag{3}$$

for some regularization parameter $\lambda > 0$.

**Discriminator updates:**  We assume simple gradient ascent of the MMD metric (3):

$$\boldsymbol{\theta}^{k+1} = \boldsymbol{\theta}^k + \eta_d \left( \sum_i p_i a(\boldsymbol{x}_i) - \sum_j \widetilde{p}_j a(\widetilde{\boldsymbol{x}}_j) - \lambda \boldsymbol{\theta}^k \right), \tag{4}$$

where $\eta_d > 0$ is the discriminator step-size. If we let $f^k(\boldsymbol{x}) := f(\boldsymbol{x}, \theta^k)$, then for any fixed $\boldsymbol{x}$, we have

$$f^{k+1}(\boldsymbol{x}) = f^k(\boldsymbol{x}) + \eta_d \left( \sum_{i=1}^{N_r} p_i K(\boldsymbol{x}, \boldsymbol{x}_i) - \sum_{j=1}^{N_g} \widetilde{p}_j K(\boldsymbol{x}, \widetilde{\boldsymbol{x}}_j) - \lambda f^k(\boldsymbol{x}) \right), \tag{5}$$

where $K(\boldsymbol{x}, \boldsymbol{x}')$ is the kernel:

$$K(\boldsymbol{x}, \boldsymbol{x}') := a(\boldsymbol{x})^\mathsf{T} a(\boldsymbol{x}'). \tag{6}$$

To make the analysis concrete, some of our results will apply to the radial basis function (RBF) kernel

$$K(\boldsymbol{x}, \boldsymbol{x}') = e^{-\frac{1}{2\sigma^2} \|\boldsymbol{x} - \boldsymbol{x}'\|^2}, \tag{7}$$

where $\sigma > 0$ is the *kernel width*.

**Generator updates:**   We will assume the generator distribution $\mathbb{P}_g$ in (1) is directly parameterized by the point mass locations, $\widetilde{\boldsymbol{X}} = \{\widetilde{\boldsymbol{x}}_j\}_{j=1}^{N_g}$, that are updated to minimize the loss:

$$\mathcal{L}_g(\boldsymbol{\theta}, \{\widetilde{\boldsymbol{x}}_j\}_{j=1}^{N_g}) := -\sum_j \widetilde{p}_j f(\widetilde{\boldsymbol{x}}_j, \boldsymbol{\theta}). \tag{8}$$

We consider a simple gradient descent update for this loss

$$\widetilde{\boldsymbol{x}}_j^{k+1} = \widetilde{\boldsymbol{x}}_j^k + \eta_g \widetilde{p}_j \nabla f^k(\widetilde{\boldsymbol{x}}_j^k). \tag{9}$$

Together, (5) and (9) define the joint dynamics of the GAN.

**Isolated Points:**   Our final key assumption is that the true samples are separated far enough so that there exists a non-empty *isolated neighborhood* $V_i$ around each sample $\boldsymbol{x}_i$ such that,

$$K(\boldsymbol{x}, \boldsymbol{x}') = 0 \text{ for all } \boldsymbol{x} \in V_i \text{ and } \boldsymbol{x}' \in V_j \text{ for all } i \neq j \tag{10}$$

In other words, the samples are separated sufficiently far apart such that they are outside the width of the kernel evaluated at another sample. See Figure 1 for an illustration.

The assumption (10) is obviously strict and is an idealization of what occurs in practice. The supplementary material discusses modifications of the results to the case where $|K(\boldsymbol{x}, \boldsymbol{x}')| \leq \epsilon$ for some small $\epsilon$ and all $\boldsymbol{x} \in V_i$ and $\boldsymbol{x}' \in V_j$.

Now consider an isolated region $V_i$ around a true point $\boldsymbol{x}_i$. At each training step $k$, let $N_i^k$ be the set of indices $j$ such that the generated points $\widetilde{\boldsymbol{x}}_j^k \in V_i$. We further suppose that $N_i^k$ is constant over time, so the points $\widetilde{\boldsymbol{x}}_j^k$ do not enter or exit this region. Then, their dynamics within the region are given by:

$$f^{k+1}(\boldsymbol{x}) = f^k(\boldsymbol{x}) + \eta_d \left( p_i K(\boldsymbol{x}, \boldsymbol{x}_i) - \sum_{j \in N_i} \widetilde{p}_j K(\boldsymbol{x}, \widetilde{\boldsymbol{x}}_j^k) - \lambda f^k(\boldsymbol{x}) \right) \qquad \forall \boldsymbol{x} \in V_i, \tag{11a}$$

$$\widetilde{\boldsymbol{x}}_j^{k+1} = \widetilde{\boldsymbol{x}}_j^k + \eta_g \widetilde{p}_j \nabla f^{k+1}(\widetilde{\boldsymbol{x}}_j^k) \qquad \forall j \in N_i \tag{11b}$$

We will call the updates (11) the *dynamical system in the region $V_i$*.

We may also choose to write the updates (11) in terms of the components of $\boldsymbol{\theta}$. Let $\boldsymbol{\theta}_i := \boldsymbol{P}_i \boldsymbol{\theta}$ where $\boldsymbol{P}_i$ is the projection onto the range space of $a(\boldsymbol{x})$ for $\boldsymbol{x} \in V_i$. Then, it can be verified that the updates in (11) can be written as:

$$\boldsymbol{\theta}_i^{k+1} = \boldsymbol{\theta}_i^k + \eta_d \left( p_i a(\boldsymbol{x}_i) - \sum_{j \in N_i} \widetilde{p}_j a(\widetilde{\boldsymbol{x}}_j^k) - \lambda \boldsymbol{\theta}_i^k(\boldsymbol{x}) \right) \tag{12a}$$

$$\widetilde{\boldsymbol{x}}_j^{k+1} = \widetilde{\boldsymbol{x}}_j^k + \eta_g \widetilde{p}_j \nabla f(\widetilde{\boldsymbol{x}}_j^k, \boldsymbol{\theta}_i), \quad \forall j \in N_i. \tag{12b}$$

We will use the notation

$$\widetilde{\boldsymbol{X}}_i = \{\widetilde{\boldsymbol{x}}_j, \ j \in N_i\}, \tag{13}$$

to denote the set of generated points $\widetilde{\boldsymbol{x}}_j$ in the isolated region. The state variables for the dynamics (12) can be represented by the pair $(\boldsymbol{\theta}_i, \widetilde{\boldsymbol{X}}_i)$. Note that we allow separate learning rates for the two update equations. Our convergence result in Theorem 1 rely on the ratio of these learning rates.

# 3 Behavior Near the True Point

Consider an isolated region $V_i$ around some true point $\boldsymbol{x}_i$. We first analyze the dynamics where

$$\widetilde{\boldsymbol{x}}_j^k \approx \boldsymbol{x}_i \ \forall j \in N_i. \tag{14}$$

That is, all the generated points in $V_i$ are close to the true point. We follow a standard linearization analysis used in several other GAN works such as [6, 12, 18]. However, a critical difference in our model is that the generated mass may not equal the true mass in a particular isolated region. We thus define the *probability mass difference*

$$\Delta_i := p_i - \sum_{j \in N_i} \widetilde{p}_j, \tag{15}$$

which represents the difference the true probability mass in the region, $p_i$, and total probability mass of the generated points in that region. Note that this parameter is missing from the analysis of Dirac-GAN [20] since $\Delta_i = 0$ trivially when only one true point exists. Several numerical behaviors of GANs emerge due to $\Delta_i \neq 0$.

**Assumption 1.** The kernel $K(\boldsymbol{x}, \boldsymbol{x}')$ is smooth and, at each true point $\boldsymbol{x}_i$:

$$\left. \frac{\partial K(\boldsymbol{x}, \boldsymbol{x}_i)}{\partial \boldsymbol{x}} \right|_{\boldsymbol{x}=\boldsymbol{x}_i} = \boldsymbol{0}, \tag{16}$$

and

$$-\left. \frac{\partial^2 K(\boldsymbol{x}, \boldsymbol{x}_i)}{\partial \boldsymbol{x}^2} \right|_{\boldsymbol{x}=\boldsymbol{x}_i} \in [k_1, k_2]\boldsymbol{I}, \quad \left. \frac{\partial^2 K(\boldsymbol{x}, \boldsymbol{x}')}{\partial \boldsymbol{x} \partial \boldsymbol{x}'} \right|_{\boldsymbol{x}=\boldsymbol{x}'=\boldsymbol{x}_i} \in [k_3, k_4]\boldsymbol{I}, \tag{17}$$

for some $k_1, k_2, k_3, k_4 > 0$ where we use the notation that $\boldsymbol{Q} \in [q_1, q_2]\boldsymbol{I}$ to mean $q_1\boldsymbol{I} \preceq \boldsymbol{Q} \preceq q_2\boldsymbol{I}$.

The condition is mild and simply requires that the kernel $K(\boldsymbol{x}, \boldsymbol{x}_i)$ has a local maxima at $\boldsymbol{x} = \boldsymbol{x}_i$ with negative curvature, and also satisfies a strict positivity condition. The assumption, for example, is satisfied by the RBF kernel (7) with $k_1 = k_2 = k_3 = k_4 = 1/\sigma^2$.

**Theorem 1.** *Fix any isolated region $V_i$, suppose the kernel satisfies Assumption 1 and let*

$$\widetilde{\boldsymbol{X}}_i^* = \{\widetilde{\boldsymbol{x}}_j^*, \ j \in N_i\}, \quad \widetilde{\boldsymbol{x}}_j^* = \boldsymbol{x}_i. \tag{18}$$

*Then, there is a unique $\boldsymbol{\theta}_i^*$ such that $(\boldsymbol{\theta}_i^*, \widetilde{\boldsymbol{X}}_i^*)$ is an equilibrium point of the GAN dynamics (12) in the region $V_i$. In addition, if the parameters $k_i$, $\lambda$ and $\mu = \eta_g/\eta_d$ are fixed, then, for sufficiently small step size $\eta_d$:*

*(a) If $\Delta_i > 0$, the equilibrium point is locally stable.*

*(b) If $\Delta_i < 0$ and $|N_i| \geq 1$, the equilibrium point is locally unstable.*

*(c) If $|N_i| = 1$ and*

$$\mu \Delta_i k_1 \widetilde{p}_1 + \min\{\lambda^2, \mu \widetilde{p}_1^2 k_3\} > 0, \tag{19}$$

*the equilibrium point is locally stable.*

*(d) If $|N_i| = 1$ and*

$$\mu \Delta_i k_2 \widetilde{p}_1 + \min\{\lambda^2, \mu \widetilde{p}_1^2 k_4\} < 0, \tag{20}$$

*the equilibrium point is locally unstable.*

*In cases (c) and (d), $\widetilde{p}_1$ denotes the point mass of the single generated point in region.*

The theorem provides simple necessary and sufficient conditions on the stability of equilibrium points at the true point. In case (a), we see that when the probability mass difference, $\Delta_i > 0$, the generated points will converge locally to the true point. That is, the generated points will converge to the true point as long as the the true point mass exceeds the total generated mass. Also, (19) will always be satisfied when $\Delta_i = 0$. So, when $|N_i| = 1$ (i.e., there is a single generated point), the generated point will converge to the true point when the generated and true point have the same mass. Examples of these convergence situations are shown in Figure 1, in $V_1$ (where $\Delta_1 = 0$) and $V_3$ (where $\Delta_3 > 0$).

Conversely, case (b) shows that when $\Delta_i < 0$ (i.e. the generated probability exceeds the true probability), the generated mass can no longer stably settle in the true point. This case is shown in $V_2$ in Fig. 1 where the generated points are repelled from the true points.

**Single vs. multiple points:** Note there is a slight difference between the cases when $|N_i| = 1$ (i.e., there is a single generated point in the region), and $|N_i| > 1$, when there is more than one generated point. As the proof of the theorem shows, when $|N_i| > 1$, the mean of the generated points may converge to the true point, but the system may have other modes that are unstable in other directions.

**Boundary cases:** Our theorem does not discuss certain boundary cases. For example, when $\Delta_i = 0$ and $|N_i| > 1$, the theorem does not state whether the equilibrium point is stable or unstable. These boundary cases are standard in linearization analyses when the eigenvalues of the linearized system are only critically stable. Critically stable linear systems will have limit cycles [28]. The stability of general nonlinear systems will be determined by the higher order dynamics, which, in our case, would be determined by the higher order terms of the kernel. However, standard non-linear systems theory results such as [28] show that any such points cannot be exponentially stable. Hence, if the points locally converge, the convergence rate may be slow.

**Relation to Dirac-GAN:** Note that our theorem recovers the stability result [20, Theorem 4.1] as a special case: When there is a single true and generated point (whereby $|N_i| = 1$ and $\Delta_i = 0$), and regularization is used ($\lambda > 0$), the theorem shows the true point is locally stable. When there is no regularization ($\lambda = 0$), the criteria in both (19) and (20) are both exactly zero, meaning the equilibrium point is not exponentially stable or unstable. In this case, [20] shows that the system has a limit cycle.

**Non-existence of exact mode collapse:** An important corollary of these results is provided by the following result.

**Corollary 1.** *Suppose the kernel satisfies Assumption 1, and the number of true and generated points are equal so that $N_r = N_g = N$ for some $N$. Also, suppose that $p_j = \widetilde{p}_j = 1/N$ for all $j$. Then, the only stable equilibrium of the generated distribution $\mathbb{P}_g$ with*

$$\mathrm{supp}(\mathbb{P}_g) \subseteq \mathrm{supp}(\mathbb{P}_r) \tag{21}$$

*is $\mathbb{P}_g = \mathbb{P}_r$.*

In Corollary 1, $\mathrm{supp}(\cdot)$ denotes the support of the distributions. So, $\mathrm{supp}(\mathbb{P}_r)$ denotes the set of non-unique values of true points $\boldsymbol{x}_i$ and and $\mathrm{supp}(\mathbb{P}_g)$ denotes the set of non-unique values of generated points $\widetilde{\boldsymbol{x}}_j$. Thus, the result rules out the possibility that the generated distribution can find a locally stable equilibrium where multiple generated points converge to a single true point.

## 4 Bad Local Minima and Approximate Mode Collapse

Mode collapse is one of the most widely-known failure mechanisms of GANs, especially for the simple GANs considered in this work [2, 11, 19, 26]. Corollary 1 appears to rule out mode collapse in the sense that there are no locally stable equilibria where multiple generated point arrive in a single true point. Indeed, Theorem 1 suggests that the generated points will be "repelled" from the true point when the generated mass exceeds the true mass, i.e., $\Delta_i < 0$. This repulsive force offers the possibility that the excess generated mass can move towards other true points.

However, in this section, we will show that even when $\Delta_i < 0$, the generated points may get stuck close to the true point. This phenomena is what we call *approximate mode collapse*.

**Theorem 2.** *Fix a region $V_i$ and consider the dynamical system* (12) *with the RBF kernel* (7)*. Assume $|N_i| > 1$ so there is more than one generated point in the region. Then, there exists a constant $c > 0$, independent of the kernel width $\sigma$, and $N_{\max}$, which is only a function of $d$, such that, if $|N_i| \leq N_{\max}$, the system has a locally stable equilibrium with $\widetilde{\boldsymbol{X}}_i^* = \{\widetilde{\boldsymbol{x}}_j^*, j \in N_i\}$ and*

$$\|\widetilde{\boldsymbol{x}}_j^* - \boldsymbol{x}_i\| \leq c\sigma, \tag{22}$$

*for all $j \in N_i$.*

The theorem states that, under certain conditions, the generated points can get stuck in a local minima, even when the generated mass exceeds the true mass. An example of this situation is illustrated in Figure 1 in the region $V_2$. The true mass (the green disk at the center of $V_2$) is itself an unstable location for the generated points since $\Delta_i < 0$. But, the points may find a stable local minima around

the true point. Moreover, the distance of the local minima to the $\boldsymbol{x}_i$ scales with the kernel width $\sigma$. Hence, for very small $\sigma$, the generated points with arbitrarily high mass may accumulate near a single true point. This phenomena can thus be seen as a type of approximate mode collapse.

The proof of Theorem 2 shows that the bad minima holds for kernels other than the RBF as well.

Prior work in mode collapse have identified at least two failure mechanisms: The first is that the discriminator may have zero gradient on the generated data when it comes from a low dimensional manifold [2]. The above result provides a constructive proof for the existence of such modes, and additionally shows that the local minima is stable, meaning the gradients push the generated data locally to the bad minima.

A second failure mechanism is catastrophic forgetting [26], that past generated values are not remembered by the discriminator. Since we use a regularized discriminator ($\lambda > 0$), our discriminator also "forgets" past values, and thus, the existence of the bad minima would be consistent with catastrophic forgetting.

# 5  Divergence

What happens to generated points isolated from all the true points $\{\boldsymbol{x}_i\}$? Consider a single generated point $\widetilde{\boldsymbol{x}}_0$ whose trajectory $\{\widetilde{\boldsymbol{x}}_0^k\}_{k \geq 0}$ satisfies

$$K(\widetilde{\boldsymbol{x}}_0^k, \widetilde{\boldsymbol{x}}_j^k) = 0 \quad \forall j \neq 0, \qquad K(\widetilde{\boldsymbol{x}}_0^k, \boldsymbol{x}_i) = 0 \quad \forall i \qquad \text{for all } k. \tag{23}$$

That is, $\widetilde{\boldsymbol{x}}_0^k$ is sufficiently far from all the other generated points and true points so that the kernel can be treated as zero. In this case, the dynamics (5) and (9) reduce to

$$f^{k+1}(\boldsymbol{x}) = (1 - \eta_d \lambda) f^k(\boldsymbol{x}) - K(\boldsymbol{x}, \widetilde{\boldsymbol{x}}_0^k) \tag{24a}$$

$$\widetilde{\boldsymbol{x}}_0^{k+1} = \widetilde{\boldsymbol{x}}_0^k + \eta_g \nabla f^k(\widetilde{\boldsymbol{x}}_0^{k+1}). \tag{24b}$$

**Theorem 3.** *Consider the dynamical system* (24) *with a translation-invariant kernel of the form* $K(\boldsymbol{x}, \boldsymbol{x}') = \phi(\|\boldsymbol{x} - \boldsymbol{x}'\|)$ *for some smooth, integrable function* $\phi(\cdot)$ *with* $\phi(0) > 0$. *Then, for any initial condition,* $\widetilde{\boldsymbol{x}}_0^0$, *and unit vector* $\boldsymbol{u} \in \mathbb{R}^d$, *and* $\lambda$ *sufficiently small, there exists an initial discriminator function* $f^0(\boldsymbol{x})$ *and velocity* $v_0 > 0$ *such that the solution to the system in equation* (24) *is*

$$\widetilde{\boldsymbol{x}}_0^k = \widetilde{\boldsymbol{x}}_0^0 + k v_0 \boldsymbol{u}. \tag{25}$$

Remarkably, the theorem shows that an isolated generated point can enter a trajectory where it continues to move linearly in a direction simply propelled by its own kernel history $K(\boldsymbol{x}, \widetilde{\boldsymbol{x}}_0^k)$. This situation is illustrated in Figure 1, where the generated point in the top right moves to the right in a straight line. We will see several examples of divergence in the numerical results as well.

# 6  Role of the Kernel Width

A key parameter of any distance based kernels, such as the RBF (7), is its *kernel width*, meaning the approximate distance at which the kernel begins to decay significantly. Our results, combined with previous analyses, provide useful insights into the role of the kernel width in GAN training.

For example, consider the simple case of the Dirac-GAN where there is a single true point $\boldsymbol{x}_0$, and single generated point $\widetilde{\boldsymbol{x}}_0$. If we fix the generated point, $\widetilde{\boldsymbol{x}}_0$, and let an RBF discriminator run to the convergence, it is known from results in [6, 12, 18] that the discriminator parameters $\boldsymbol{\theta}^k$ will converge to some values $\boldsymbol{\theta}^k \to \boldsymbol{\theta}^*$, and the resulting generator loss in (8) is given by

$$\mathcal{L}_g(\boldsymbol{\theta}^*, \widetilde{\boldsymbol{x}}_0) = \frac{1}{\lambda} \left( 1 - e^{-\|\boldsymbol{x}_0 - \widetilde{\boldsymbol{x}}_0\|^2 / 2\sigma^2} \right). \tag{26}$$

The supplementary material reviews these results in a more general setting.

Now, if the kernel width $\sigma$ is very large, the loss will have a small gradient when $\widetilde{\boldsymbol{x}}_0$ is close to $\boldsymbol{x}_0$, i.e., the generated point is close to the true value. On other hand, if $\sigma$ is very small, the loss will have a small gradient when $\widetilde{\boldsymbol{x}}_0$ is far from $\boldsymbol{x}_0$. The selection of the appropriate $\sigma$ must balance the convergence rates in these two regimes. In addition, when $\sigma$ is large, the discriminator is approximately linear and therefore cannot distinguish nonlinear regions.

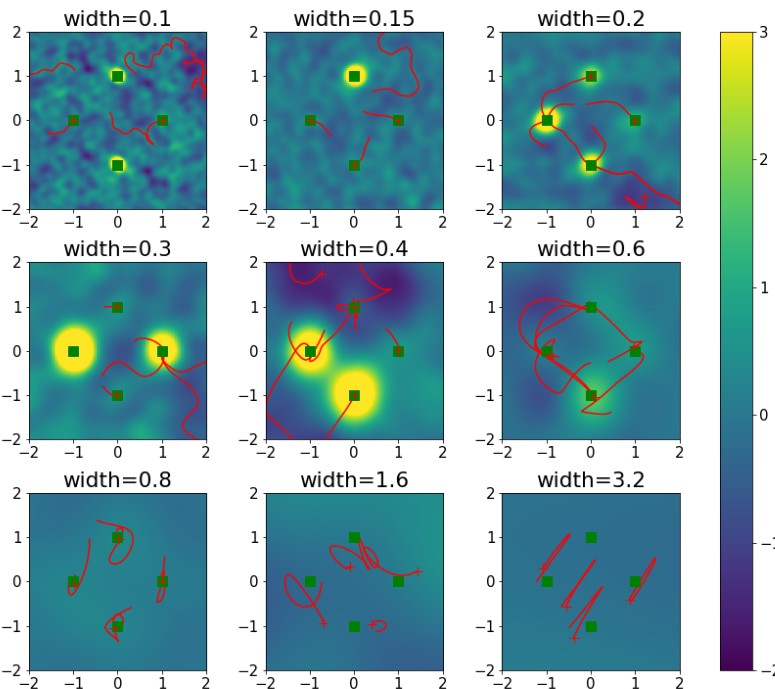

Figure 2: **Behavior of joint GAN training with kernel width**: Example trajectories of generated points over the course of training (red lines with final point marked as a cross), true distribution (green), final discriminator (blue and yellow colormap).

The results for the isolated points model provide additional insight. The isolated points assumption arises when the true points become separated at a distance much greater than $\sigma$. In such cases, we have seen that at least two failure modes become possible: (1) approximate mode collapse, as described in Theorem 2; and (2) divergence, as described in Theorem 3, where generated points isolated from all true points diverge in arbitrary directions. These two failure mechanisms may be prevented with a wider kernel width to preclude the isolated points assumption. However, wide kernel widths come at the price of poor discriminator power and slow convergence near the true point.

## 7 Numerical Examples

We conduct a simple experiment on low-dimensional, synthetic data to illustrate the behavior that the theory predicts. We also look at the frequency of GAN failure modes as a function of kernel width of the discriminator. Our main observation is that *failure most often occurs when the model is in an isolated points regime* at small kernel width.

Two simple datasets for the true data are used: A set of $N_r = 4$ points arranged on uniformly on the unit circle in dimension $d = 2$; and a set of $N_r = 10$ points randomly distributed on the unit sphere in dimension $d = 10$. In both cases, we initialize $N_g = N_r$ generated point as Gaussians with zero mean and $\mathbb{E} \|\widetilde{x}_j\|^2 = 1$. We approximate the RBF discriminator using a random Fourier feature map as in [24]. We set $\lambda = 0.01$ and $\eta_d = \eta_g = 10^{-3}$ and use 40000 training steps. Other details are in the Supplementary material.

Figure 2 show example trajectories for the $d = 2$ case for different kernel widths $\sigma$. For the two dimensional setting we visualize the trajectories of generated points over the course of training in Figure 2. We see a diverging behavior at very small kernel widths, as these generated points are not influenced by the kernels of the true points. Instead they "wander" off. Although Theorem 3 shows linear diverging trajectories, other diverging trajectories may be possible due to the randomness

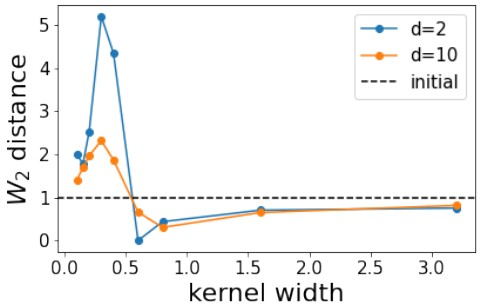
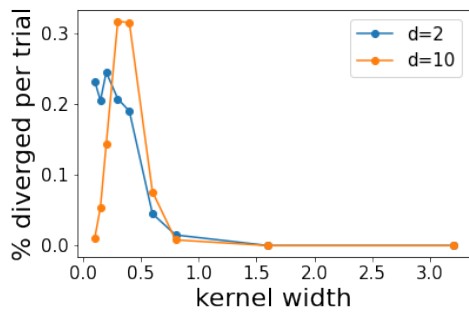

(a) Normalized Wasserstein distance          (b) Frequency of divergence

Figure 3: Frequency of different GAN failure modes in a four point, two dimensional setting and a 10 point, 10 dimensional setting. (a) The median change in Wasserstein distance between true and generated distributions after 40k iterations. Different kernel widths of an RBF kernel discriminator are tested. (b) The average percentage of diverging generated points after training a GAN with RBF kernel discriminator. A generated point is considered diverging if $\|\widetilde{x}\|_2 > 2$

in Fourier feature map. Also, in the example at $\sigma = 0.6$, we see an approximate mode collapse where two generated points converge to a single true point. Finally, at very large kernel widths the discriminator is not able to properly distinguish between distributions, and as a result we see large oscillations. The discriminator gradients are much smoother, meaning they are closer to the linear discriminator regime from the original Dirac-GAN analysis [19, 26].

To measure how these failure mechanisms affect the overall performance, for each kernel width $\sigma$, we run 100 trials of the true and generated points. Figure 3 plots the median performance of the GAN along two metrics. In Figure 3(a), we measure the Wasserstein-2 distance [14] between true and generated distributions before and after training period for a fixed number of iterations. The Wasserstein-2 distance is estimated by [7] and provides a error due to diverging points and mode collapse. We see that at very low kernel widths, the normalized error can be even greater than one, meaning that the generated points are further from the true points than their initial value. This behavior is a result of the divergence. At very high kernel widths, there is also significant error due to the slow convergence and lack of power of the discriminator.

As a simple measure of frequency of divergence, Figure 3b measures the average fraction of generated points, $\widetilde{x}_j$, where $\|\widetilde{x}\|_2 > 2$ at the end of the iterations. Since the true points are on the unit sphere ($\|x_i\| = 1$), this condition can be considered as a case where the generated points have move significantly from all the true points. As expected, we see that the frequency of divergence increases with small kernel width. However, for dimension $d = 10$, the frequency of divergence is small for very low kernel widths. This phenomena is a result of the fact that we are approximating the RBF with random features. As shown in [24], the number of features required for a good approximation to the RBF grows with $d^2$.

## 8 Potential Solutions via Multi-Scale Kernels

The above discussion and simulations shows that there is a fundamental trade-off with respect to the kernel width. On the one hand, very wide kernels tend to provide slow rates of convergence. In addition, they are not able to accurately discriminate between true points that are close. On the other hand, very narrow width kernels can result in true points being isolated from one another. The results in this paper show that isolated points can lead to both approximate mode collapse and divergence of generated points far away from the true distribution. Fortunately, the analysis in the paper suggests a way to avoid both of these conditions. Specifically, suppose we consider a kernel where the discriminator is of the form:

$$f(x, \theta_1, \theta_2) := a_1(x)^{\mathsf{T}} \theta_1 + a_2(x)^{\mathsf{T}} \theta_2, \tag{27}$$

where $a_1(x)$ and $a_2(x)$ are two basis functions and $\theta = (\theta_1, \theta_2)$ are the parameters. The kernel for this discriminator is:

$$K(x, x') = K_1(x, x') + K_2(x, x'), \tag{28}$$

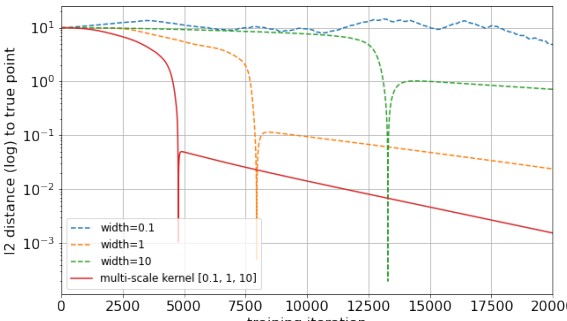

Figure 4: GAN training dynamics using fixed-width kernel discriminators (blue, yellow, and green dashed lines) compared to the GAN training dynamics using a feature map constructed by concatenation of fixed-width feature maps (illustrated by solid line). A concatenated feature map produces a kernel that is a linear combination of its component kernels

where $K_i(x, x') = a_i(x)^\intercal a_i(x')$ are the kernels for each of the bases.

Now, suppose that $a_1(x)$ and $a_2(x)$ are selected so that $K_1(x)$ has a wide width and $K_2(x)$ has a small width. We call such a kernel *multi-scale*. The overall kernel (28) will have a heavy tail component due to the "wide" kernel $K_1(x)$. Hence, it can avoid the isolated points problem. On the other hand, since $K_2(x)$ has a small width, the kernel will have a "sharp" component. This may enable fast convergence near the true distribution.

To illustrate the potential use of such a multi-scale kernel, Fig. 4 compares the performance of kernels with three fixed widths of $\sigma \in \{0.1, 1, 10\}$ with a single kernel concatenated with all three widths. The true and generated data are single point masses with an initial distance of 10. As expected, the very low width kernel ($\sigma = 0.1$) fails to converge while the wide width kernels ($\sigma = 1, 10$) converge slowly. In contrast, the concatenated kernel is able to get fast convergence in both the initial and later stages. Of course, further work will be needed to find out the best architectures for networks with multiple widths for complex practical data. In the context of neural network discriminators, such multi-scale kernel behavior may be achieved using parallel networks with varying depths, or the use of skip connections. However, designing such a discriminator requires further investigation.

## 9   Conclusion

In this paper we propose a new framework for understanding behaviors exhibited by training GANs. By assuming isolated neighborhoods around each true point, we can decouple the kernel GAN training dynamics that occur when jointly updating a generator and discriminator. Using this model we theoretically derive conditions that explain well-known training behavior in GANs. While a stable local equilibrium exists when true and generated distributions match exactly, bad local minima exist as well. Specifically a true point can "greedily" hold onto multiple generated points in a region determined by the discriminator kernel width, leading to the phenomenon of approximate mode collapse. We provide theoretical and empirical evidence of a diverging failure mode as well, where one or more generated points completely escape the influence of the true distribution and travel along arbitrary trajectories. Through this analysis, it becomes clear that kernel width in particular plays an important role in this behavior.

There are several possible lines of future work. Most importantly, we have studied a simple GAN without gradient penalties, as is commonly used in methods such as [2, 3, 13, 17]. Future work can also consider the convergence. Our linearized analysis provides the eigenvalues which can be used for rates close to the equilibrium as performed in [21, 22, 30]. Finally, there is a large body of literature connecting kernels to the neural networks via the so-called neural tangent kernel (NTK) [15] and may be useful to study here as well.

## Acknowledgments and Disclosure of Funding

S. Rangan was supported in part by NSF grants 1952180, 1925079, 1564142, 1547332, the SRC, and the industrial affiliates of NYU WIRELESS.

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
