# Appendix

## A   Review of Local Stability Results

We provide a brief review of standard definitions and stability results that we use in the theorem statements and proofs. All the material can be found in any text in nonlinear systems such as [28]. A (discrete-time) dynamical system is simply a recursion of the form

$$z^{k+1} = \Phi(z^k), \quad z^0 = z_0, \tag{29}$$

for some (possibly nonlinear) mapping $\Phi(\cdot)$. Here, $z_0$, is the initial condition. A point $z^*$ is called an *equilibrium point* of (29) if

$$z^* = \Phi(z^*). \tag{30}$$

The importance of an equilibrium point is that if a dynamical system is initialized at an equilibrium point, $z^0 = z^*$, then it will remain there: $z^k = z^*$ for all $k \geq 0$.

An equilibrium point $z^*$ is said to be *(locally) stable* if, given any $\epsilon > 0$, there exists a $\delta > 0$ such that

$$\|z^0 - z^*\| < \delta \Rightarrow \|z^k - z^*\| < \epsilon \text{ for all } k.$$

That is, the system can remain arbitrarily close to the equilibrium point if it starts sufficiently close. A system is *unstable* if it is not stable. An equilibrium point is *asymptotically stable* if it is stable and $\delta > 0$ can be chosen such that

$$\|z^0 - z^*\| < \delta \Rightarrow \lim_{k \to \infty} z^k = z^*.$$

That is, the $z^k$ will converge to $z^*$. The system is *exponentially stable*, if there exists a $\delta > 0$, $c \geq 1$, and $\rho \in [0, 1)$ such that

$$\|z^0 - z^*\| < \delta \Rightarrow \|z^k = z^*\| \leq c\rho^k \text{ for all } k.$$

Clearly, exponentially stable $\Rightarrow$ asymptotically stable $\Rightarrow$ stable.

The system (29) is *linear* if $\Phi(z) = Az$ for some matrix $A$. For a linear system, $z^* = 0$ is always an equilibrium point. The stability of $z^* = 0$ is completely determined by the eigenvalues of $A$. Specifically, if we let $\mathrm{spec}(A)$ to denote the set of eigenvalues of a matrix $A$, then we have the following well-known result:

**Lemma 1** ( [28]). *For a linear dynamical system (29) with $\Phi(z) = Az$ for some matrix $A$:*

   (a) $z^* = 0$ is exponentially stable iff $|\rho| < 1$ for all $\rho \in \mathrm{spec}(A)$.

   (b) $z^* = 0$ is stable iff $|\rho| \leq 1$ for all $\rho \in \mathrm{spec}(A)$.

   (c) If there exists a single $\rho \in \mathrm{spec}(A)$ with $|\rho| > 1$, then $z^* = 0$ is unstable.

For non-linear systems, the local stability can be determined by the eigenvalues of the Jacobian of $\Phi(z^*)$, which we will denote by $\Gamma(z^*)$:

$$\Gamma(z^*) := \mathrm{spec}\left(\frac{\partial \Phi(z^*)}{\partial z}\right). \tag{31}$$

**Lemma 2** ( [28]). *Consider a dynamical system (29) for some smooth function $\Phi(z)$. Let $z^*$ be a equilibrium point, define $\Gamma(z^*)$ as in (31), the spectrum of the Jacobian of $\Phi(z)$ at $z = z^*$. Then:*

   *(a) $z^*$ is exponentially stable iff $|\rho| < 1$ for all $\rho \in \Gamma(z^*)$.*

   *(b) If there exists a single $\rho \in \Gamma(z^*)$ with $|\rho| > 1$, then $z^*$ is locally unstable.*

A key difference with non-linear systems is that when the max modulus eigenvalue is on the unit circle (i.e., $\max |\rho| = 1$), then system may be unstable or stable. The stability will be determined by higher-order terms.

# B Restricted MMD Distance

As preparation for the proofs, we next introduce a key function that we will call the restricted maximum mean discrepancy (MMD) distance. This concept is a specialization of the MMD distance in [6, 12, 18] for the local dynamical system (12).

Fix the generator locations

$$\widetilde{\boldsymbol{X}} = \{\widetilde{\boldsymbol{x}}_j, j = 1, \ldots, N_g\}, \tag{32}$$

and consider the corresponding generated distribution $\mathbb{P}_g = \sum_j \widetilde{p}_j \delta(\boldsymbol{x} - \widetilde{\boldsymbol{x}}_j)$. Suppose we run the discriminator update in (4) to convergence so that $\boldsymbol{\theta}^k \to \boldsymbol{\theta}_i^*$ for some $\boldsymbol{\theta}^*$. A well-known result of GANs is that the resulting generator loss (8) is given by

$$\mathcal{L}_g(\boldsymbol{\theta}^*, \widetilde{\boldsymbol{X}}) = \frac{1}{2\lambda} \| \mathbb{P}_r - \mathbb{P}_g \|_K^2, \tag{33}$$

where the norm is the so-called squared *kernel maximum mean discrepancy* (MMD) distance [6,12,18]

$$\| \mathbb{P}_r - \mathbb{P}_g \|_K^2 = \mathbb{E}_{\boldsymbol{x},\boldsymbol{x}' \sim \mathbb{P}_r} K(\boldsymbol{x}, \boldsymbol{x}') - 2\mathbb{E}_{\boldsymbol{x} \sim \mathbb{P}_r, \widetilde{\boldsymbol{x}} \sim \mathbb{P}_g} K(\boldsymbol{x}, \widetilde{\boldsymbol{x}}) + \mathbb{E}_{\widetilde{\boldsymbol{x}}, \widetilde{\boldsymbol{x}}' \sim \mathbb{P}_g} K(\widetilde{\boldsymbol{x}}, \widetilde{\boldsymbol{x}}') \tag{34}$$

For the discrete distributions (1), the squared MMD distance (33) can be written as a function of the point mass locations

$$\| \mathbb{P}_r - \mathbb{P}_g \|_K^2 = 2J(\widetilde{\boldsymbol{X}}), \tag{35}$$

where

$$J(\widetilde{\boldsymbol{X}}) := \frac{1}{2} \sum_{i,k} p_i p_k K(\boldsymbol{x}_i, \boldsymbol{x}_k) - \sum_{i,j} p_i \widetilde{p}_j K(\boldsymbol{x}_i, \widetilde{\boldsymbol{x}}_j) + \frac{1}{2} \sum_{j,k} \widetilde{p}_j \widetilde{p}_k K(\boldsymbol{x}_j, \widetilde{\boldsymbol{x}}_k). \tag{36}$$

In (36), we have omitted the dependence on the locations of the true distributions $\boldsymbol{x}_i$ as well as the true and generated weights, $p_i$ and $\widetilde{p}_j$, since these are fixed in our model.

To analyze the dynamics in a isolated region $V_i$, we define the *restricted squared kernel MMD distance* as the function

$$J_i(\widetilde{\boldsymbol{X}}_i) := \frac{1}{2} p_i^2 K(\boldsymbol{x}_i, \boldsymbol{x}_i) - \sum_{j \in N_i} p_i \widetilde{p}_j K(\boldsymbol{x}_i, \widetilde{\boldsymbol{x}}_j) + \frac{1}{2} \sum_{j,k \in N_i} \widetilde{p}_j \widetilde{p}_k K(\widetilde{\boldsymbol{x}}_j, \widetilde{\boldsymbol{x}}_k). \tag{37}$$

This function is the squared kernel MMD distance (36), but only containing the terms with the single true point $\boldsymbol{x}_i$ and the set of generated points $\widetilde{\boldsymbol{X}}_i = \{\widetilde{\boldsymbol{x}}_j, j \in N_i\}$ in the isolated region $V_i$.

Similar to the MMD analysis in [6, 12, 18], we show that the critical points of the restricted kernel squared MMD distance are equilibrium points of the local dynamics (12). To state the result, let $\widetilde{\boldsymbol{X}}_i^* = \{\widetilde{\boldsymbol{x}}_j^*, j \in N_i\}$ be a critical point of $J_i(\widetilde{\boldsymbol{X}}_i)$ in (37) meaning

$$\frac{\partial J_i(\widetilde{\boldsymbol{X}}_i)}{\partial \widetilde{\boldsymbol{x}}_j} \bigg|_{\widetilde{\boldsymbol{X}}_i = \widetilde{\boldsymbol{X}}_i^*} = \boldsymbol{0}, \quad \text{for all } j \in N_i. \tag{38}$$

Let

$$\boldsymbol{\theta}_i^* := \frac{1}{\lambda} \left[ p_i a(\boldsymbol{x}_i) - \sum_j \widetilde{p}_j a(\widetilde{\boldsymbol{x}}_j^*) \right] \tag{39a}$$

$$f^*(\boldsymbol{x}) := f(\boldsymbol{x}, \boldsymbol{\theta}_i^*) = \frac{1}{\lambda} \left[ p_i K(\boldsymbol{x}, \boldsymbol{x}_i) - \sum_j \widetilde{p}_j K(\boldsymbol{x}, \widetilde{\boldsymbol{x}}_j^*) \right]. \tag{39b}$$

The following is similar to the results in [6, 12, 18], but applied to the restricted squared MMD distance.

**Lemma 3.** *Let $\widetilde{\boldsymbol{X}}_i^*$ be a critical point of the restricted squared MMD distance $J_i(\widetilde{\boldsymbol{X}}_i)$ for some $i$. That is, $\widetilde{\boldsymbol{X}}_i^*$ satisfies (38). Define $\boldsymbol{\theta}_i^*$ as in (39a). Then, the pair $(\boldsymbol{\theta}_i^*, \widetilde{\boldsymbol{X}}_i^*)$ is an equilibrium point of the dynamics (12) in the isolated region $V_i$. Conversely, if $(\boldsymbol{\theta}_i^*, \widetilde{\boldsymbol{X}}_i^*)$ is an equilibrium point of*

the dynamics (12), then $\widetilde{\boldsymbol{X}}_i^*$ is a critical point of $J_i(\widetilde{\boldsymbol{X}}_i)$. In addition, at any critical point, $\widetilde{\boldsymbol{X}}_i^*$, of $J_i(\widetilde{\boldsymbol{X}}_i)$,

$$\left.\frac{\partial^2 J_i(\widetilde{\boldsymbol{X}}_i)}{\partial \widetilde{\boldsymbol{x}}_j^2}\right|_{\widetilde{\boldsymbol{X}}_i = \widetilde{\boldsymbol{X}}_i^*} = -\lambda H(\widetilde{\boldsymbol{x}}_j^*, \boldsymbol{\theta}_i^*), \tag{40}$$

where $H(\widetilde{\boldsymbol{x}}, \boldsymbol{\theta}_i)$ is the Hessian of the discriminator

$$H(\boldsymbol{x}, \boldsymbol{\theta}_i) := \frac{\partial^2 f(\boldsymbol{x}, \boldsymbol{\theta}_i)}{\partial \boldsymbol{x}^2}. \tag{41}$$

*Proof.* First suppose that $\widetilde{\boldsymbol{X}}_i^*$ is a critical point of $J_i(\widetilde{\boldsymbol{X}}_i)$ and $\boldsymbol{\theta}_i^*$ as in (39a). We need to show that $(\boldsymbol{\theta}_i^*, \widetilde{\boldsymbol{X}}_i^*)$ are fixed points of (12). That is, we need to show:

$$p_i a(\boldsymbol{x}_i) - \sum_{j \in N_i} \widetilde{p}_j a(\widetilde{\boldsymbol{x}}_j^*) - \lambda \boldsymbol{\theta}_i^* = \boldsymbol{0} \tag{42a}$$

$$\nabla f^*(\widetilde{\boldsymbol{x}}_j^*) = \boldsymbol{0}. \tag{42b}$$

From (39a), we have

$$p_i a(\boldsymbol{x}_i) - \sum_j \widetilde{p}_j a(\widetilde{\boldsymbol{x}}_j^*) - \lambda \boldsymbol{\theta}_i^* = \boldsymbol{0},$$

which proves (42a). Also, the partial derivative of of $J_i(\widetilde{\boldsymbol{X}}_i)$ in (37) is

$$\frac{\partial J_i(\widetilde{\boldsymbol{X}}_i)}{\partial \widetilde{\boldsymbol{x}}_j} = -p_i \frac{\partial K(\widetilde{\boldsymbol{x}}_j, \boldsymbol{x}_i)}{\partial \widetilde{\boldsymbol{x}}_j} + \sum_j \widetilde{p}_j \frac{\partial K(\widetilde{\boldsymbol{x}}_j, \widetilde{\boldsymbol{x}}_k)}{\partial \widetilde{\boldsymbol{x}}_j} = -\lambda \nabla f^*(\widetilde{\boldsymbol{x}}_j), \tag{43}$$

where, in the last step, we used the definition of $f^*(\boldsymbol{x})$ in (39b). Since $\widetilde{\boldsymbol{X}}_i^*$ is a local minima of $J_i(\widetilde{\boldsymbol{X}}_i)$ we have

$$\nabla f^*(\widetilde{\boldsymbol{x}}_j^*) = \left.\frac{\partial J_i(\widetilde{\boldsymbol{X}}_i)}{\partial \widetilde{\boldsymbol{x}}_j}\right|_{\widetilde{\boldsymbol{x}}_j = \widetilde{\boldsymbol{x}}_j^*} = \boldsymbol{0},$$

which shows (42b). The converse is proven by reversing the above steps. That is, if $(\boldsymbol{\theta}_i^*, \widetilde{\boldsymbol{X}}_i^*)$ are equilibrium points of (12), then $\widetilde{\boldsymbol{X}}_i^*$ is a critical point of $J_i(\widetilde{\boldsymbol{X}}_i)$. In addition, taking the derivative of (43) shows (40). $\square$

To analyze the stability of the equilibrium point in Lemma 3, we now apply the linearization methods reviewed in Appendix A. As mentioned in the introduction, most of the stability results for GANs follow a similar procedure. To simplify the notation, WLOG assume that the set of indices $j \in N_i$ are

$$N_i = \{1, \ldots, N\}, \tag{44}$$

so the the set of points $N_i$ are simply the first $N$ generated points for some $N$. Let $\boldsymbol{z}^k$ denote the state variables

$$\boldsymbol{z}^k := (\boldsymbol{\theta}_i^k, \widetilde{\boldsymbol{X}}_i^k) = (\boldsymbol{\theta}_i^k, \widetilde{\boldsymbol{x}}_1^k, \ldots, \widetilde{\boldsymbol{x}}_N^k) \tag{45}$$

for the dynamics (12). We can write these updates as

$$\boldsymbol{z}^{k+1} = \Phi(\boldsymbol{z}^k), \tag{46}$$

for some non-linear function $\Phi(\cdot)$. Then $\boldsymbol{z}^* = (\boldsymbol{\theta}_i^*, \widetilde{\boldsymbol{X}}_i^*)$ is an equilibrium point of (12) if and only if $\boldsymbol{z}^* = \Phi(\boldsymbol{z}^*)$.

To apply Lemma 2, the following lemma will allow us to compute the eigenvalues of the Jacobian of the linearization.

**Lemma 4.** *Let $\boldsymbol{z}^* = (\boldsymbol{\theta}_i^*, \widetilde{\boldsymbol{X}}_i^*)$ be an equilibrium point as in Lemma 3 and let $\Gamma(\boldsymbol{z}^*)$ be the spectrum of the Jacobian of the update map. Then $\rho \in \Gamma(\boldsymbol{z}^*)$ if and only if $\rho$ is of the form*

$$\rho = 1 + \eta_d s, \tag{47}$$

*where $s = -\lambda$ or $s$ is a root of the polynomial*

$$\psi(s) = \det(\boldsymbol{D}(s)), \quad \boldsymbol{D}(s) := (s+\lambda)(s\boldsymbol{I} + \boldsymbol{Q}) + \boldsymbol{R}, \tag{48}$$

*where $\boldsymbol{Q}$ and $\boldsymbol{R}$ are the block matrices with components*

$$\boldsymbol{Q}_{ij} = -\mu\widetilde{p}_i H(\widetilde{\boldsymbol{x}}_j^*, \theta^*)\delta_{ij}, \quad \boldsymbol{R}_{ij} = \mu\widetilde{p}_i\widetilde{p}_j\frac{\partial^2}{\partial\boldsymbol{x}\partial\boldsymbol{x}'} \, K(\boldsymbol{x}, \boldsymbol{x}')|_{\boldsymbol{x}=\widetilde{\boldsymbol{x}}_i^*, \boldsymbol{x}'=\widetilde{\boldsymbol{x}}_j^*}. \tag{49}$$

*Proof.* The update map $\boldsymbol{z}^{k+1} = \Phi(\boldsymbol{z}^k)$ is defined by the equations (12). Let $\boldsymbol{F}$ be its Jacobian evaluated at $\boldsymbol{z} = \boldsymbol{z}^*$:

$$\boldsymbol{F} = \frac{\partial\Phi(\boldsymbol{z}^*)}{\partial\boldsymbol{z}}. \tag{50}$$

Conformal with the components of $\boldsymbol{z}$ in (45), the Jacobian is given by

$$\boldsymbol{F} = \boldsymbol{I} + \eta_d\boldsymbol{A}, \tag{51}$$

where

$$\boldsymbol{A} = \begin{bmatrix} A_{00} & A_{01} & \cdots & A_{0N} \\ A_{10} & A_{10} & \cdots & A_{1N} \\ A_{N0} & A_{N0} & \cdots & A_{NN} \end{bmatrix} \tag{52}$$

and

$$A_{00} = -\lambda\boldsymbol{I}, \tag{53a}$$
$$A_{0j} = -\widetilde{p}_j G(\widetilde{\boldsymbol{x}}_j^*) \tag{53b}$$
$$A_{j0} = \mu\widetilde{p}_j G(\widetilde{\boldsymbol{x}}_j^*)^{\mathsf{T}} \tag{53c}$$
$$A_{jk} = \mu\widetilde{p}_j H(\widetilde{\boldsymbol{x}}_j^*, \boldsymbol{\theta}_i^*)\delta_{jk}, \tag{53d}$$

where $G(\boldsymbol{x})$ is the gradient of the basis functions

$$G(\boldsymbol{x}) := \frac{\partial a(\boldsymbol{x})}{\partial\boldsymbol{x}}, \tag{54}$$

and $H(\boldsymbol{x}, \boldsymbol{\theta}_i)$ is the Hessian in (41).

The matrix $\boldsymbol{A}$ in (52) can in turn be written as

$$\boldsymbol{A} = \begin{bmatrix} -\lambda\boldsymbol{I}_p & -\boldsymbol{GP} \\ \mu\boldsymbol{PG}^{\mathsf{T}} & -\boldsymbol{Q} \end{bmatrix} \tag{55}$$

where

$$\boldsymbol{P} = \mathrm{diag}(\widetilde{p}_1\boldsymbol{I}_d, \cdots, \widetilde{p}_N\boldsymbol{I}_d), \tag{56a}$$
$$\boldsymbol{G} = [G(\widetilde{\boldsymbol{x}}_1^*), \dots, G(\widetilde{\boldsymbol{x}}_N^*)], \tag{56b}$$
$$\boldsymbol{Q} = -\mu\mathrm{diag}(\widetilde{p}_1 H(\widetilde{\boldsymbol{x}}_1, \boldsymbol{\theta}_i^*), \cdots, \widetilde{p}_N H(\widetilde{\boldsymbol{x}}_N, \boldsymbol{\theta}_i^*)), \tag{56c}$$

and $p$ is the dimension of $\boldsymbol{\theta}_i$ and $d$ is the dimension of $\boldsymbol{x}$. Note that the matrix $\boldsymbol{P}$ and $\boldsymbol{Q}$ have dimnesions $Nd \times Nd$. Hence for any $s$,

$$s\boldsymbol{I} - \boldsymbol{A} = \begin{bmatrix} (s+\lambda)\boldsymbol{I}_p & \boldsymbol{GP} \\ -\mu\boldsymbol{PG}^{\mathsf{T}} & s\boldsymbol{I}_{Nd} + \boldsymbol{Q} \end{bmatrix}. \tag{57}$$

Using the determinant of the Schur complement,

$$\det(s\boldsymbol{I} - \boldsymbol{A}) \overset{(a)}{=} \det((s+\lambda)\boldsymbol{I}_p)\det\left(s\boldsymbol{I}_{Nd} + \boldsymbol{Q} + \frac{1}{s+\lambda}\boldsymbol{R}\right)$$

$$\overset{(b)}{=} (s+\lambda)^{p-Nd}\det\left((s+\lambda)(s\boldsymbol{I}_{Nd} + \boldsymbol{Q}) + \boldsymbol{R}\right) \overset{(c)}{=} (s+\lambda)^{p-Nd}\psi(s), \tag{58}$$

where in step (a), we define $\boldsymbol{R}$ as

$$\boldsymbol{R} = \mu\boldsymbol{PG}^{\mathsf{T}}\boldsymbol{GP}, \tag{59}$$

in step (b), we use the property that $\det(\alpha\boldsymbol{M}) = \alpha^m\det(\boldsymbol{M})$ for any $\boldsymbol{M} \in \mathbb{C}^{m \times m}$; and in step (c), we used the definition of $\psi(s)$ in (48). This proves that the eigenvalues of $\boldsymbol{A}$ are either $s = -\lambda$ or

the roots of $\psi(s)$. Finally, note that the definition of $\boldsymbol{Q}$ in (56c) agrees with $\boldsymbol{Q}$ in (49). Also, the components of $\boldsymbol{R}$ in (59) are

$$\boldsymbol{R}_{jk} = \mu[\boldsymbol{P}\boldsymbol{G}^{\mathsf{T}}\boldsymbol{G}\boldsymbol{P}]_{jk} = \mu\widetilde{p}_j\widetilde{p}_k G(\widetilde{\boldsymbol{x}}_j^*)G(\widetilde{\boldsymbol{x}}_k^*) = \mu\widetilde{p}_j\widetilde{p}_k \left. \frac{\partial^2}{\partial\boldsymbol{x}\partial\boldsymbol{x}'}K(\boldsymbol{x},\boldsymbol{x}')\right|_{\boldsymbol{x}=\boldsymbol{x}_j^*,\boldsymbol{x}'=\boldsymbol{x}_k^*}, \tag{60}$$

where in the last step we used the definition of $G(\boldsymbol{x})$ in (54) and the fact that the kernel is $K(\boldsymbol{x},\boldsymbol{x}') = a(\boldsymbol{x})^{\mathsf{T}}a(\boldsymbol{x}')$. $\qquad\square$

Combining Lemma 2 and Lemma 4, we obtain the following simple stability test.

**Lemma 5.** *Let $\boldsymbol{z}^* = (\boldsymbol{\theta}_i^*, \widetilde{\boldsymbol{X}}_i^*)$ be an equilibrium point as in Lemma 3. Consider the real roots, $\alpha$ for $\psi(\alpha) = 0$, where $\psi(\alpha)$ is the polynomial in in (48). If all for real roots, $\alpha < 0$, the equilibrium point is locally stable for sufficiently small $\eta_d$. Conversely, if there is a single positive real root, $\alpha > 0$, the equilibrium point is locally unstable for all $\eta_d$ sufficiently small.*

*Proof.* First suppose that all real roots $\alpha$ of $\psi(\alpha) = 0$ are negative, i.e., $\alpha < 0$. Now suppose that $s = \alpha + i\beta$ be a, possibly complex, root of the $\psi(s)$. Then, there exists a vector $\boldsymbol{v} \neq \boldsymbol{0}$ with

$$\boldsymbol{D}(s)\boldsymbol{v} = ((s+\lambda)(s\boldsymbol{I}+\boldsymbol{Q})+\boldsymbol{R})\boldsymbol{v} = \boldsymbol{0}, \tag{61}$$

whree $\boldsymbol{D}(s)$ is defined in (48). Since $\boldsymbol{Q} = \boldsymbol{Q}^{\mathsf{T}}$ and $\boldsymbol{R} = \boldsymbol{R}^{\mathsf{T}}$, it follows that

$$\boldsymbol{D}(\alpha)\boldsymbol{v} = ((\alpha+\lambda)(\alpha\boldsymbol{I}+\boldsymbol{Q})+\boldsymbol{R})\boldsymbol{v} = \boldsymbol{0}. \tag{62}$$

Hence, $\alpha$ is a root of $\psi(\alpha) = 0$. From the assumption, $\alpha < 0$. Hence, the magnitude squared of the eigenvalue $\rho$ in (47) is

$$|\rho|^2 = (1+\eta_d\alpha)^2 + (\eta_d\beta)^2 = 1 + 2\alpha\eta_d + \eta_d^2(\alpha^2 + \beta^2) \tag{63}$$

Since $\alpha < 0$, we will have $|\rho|^2 < 1$ for all $\eta_d$ with

$$\eta_d < \ \min \ \frac{-\alpha}{(\alpha^2+\beta^2)}, \tag{64}$$

where the minimum is over all roots of $\psi(s) = 0$ with $s = \alpha + i\beta$.

Conversely, suppose that there is at least one real root $\alpha > 0$ with $\psi(\alpha) = 0$. The magnitude squared of the corresponding eigenvalue $\rho$ in (47) is

$$|\rho|^2 = (1+\eta_d\alpha)^2 = 1 + 2\alpha\eta_d + \eta_d^2\alpha^2. \tag{65}$$

We will have $|\rho| > 1$ for $\eta_d < 1/\alpha$. $\qquad\square$

## C  Proof of Theorem 1

Taking the derivatives of $J_i(\widetilde{\boldsymbol{X}}_i)$ in (37) at $\widetilde{\boldsymbol{x}}_j = \boldsymbol{x}_i$:

$$\left.\frac{\partial}{\partial\widetilde{\boldsymbol{x}}_j} \ J_i(\widetilde{\boldsymbol{x}})\right|_{\widetilde{\boldsymbol{x}}_j=\boldsymbol{x}_i} = -p_i \left.\frac{\partial K(\boldsymbol{x},\boldsymbol{x}_i)}{\partial\boldsymbol{x}}\right|_{\boldsymbol{x}=\boldsymbol{x}_i} + \sum_j \widetilde{p}_j \left.\frac{\partial K(\boldsymbol{x},\boldsymbol{x}_i)}{\partial\boldsymbol{x}}\right|_{\boldsymbol{x}=\boldsymbol{x}_i} = \boldsymbol{0}, \tag{66}$$

where the final step uses the assumption (16). Thus, the points $\widetilde{\boldsymbol{x}}_j^* = \boldsymbol{x}_i$ are critical points of $J_i(\widetilde{\boldsymbol{x}})$ and, by Lemma 3, they are equilibrium points of (12).

We next apply Lemma 5 to determine the stability of the equilibrium points. Corresponding to the equilibrium points $\widetilde{\boldsymbol{x}}_j^* = \boldsymbol{x}_i$, the discriminator in (39b) is

$$f^*(\boldsymbol{x}) = \frac{\Delta_i}{\lambda}K(\boldsymbol{x},\boldsymbol{x}_i), \tag{67}$$

where $\Delta_i$ is defined in (15). Hence, the Hessian of the discriminator in (41) at $\boldsymbol{x} = \boldsymbol{x}_i$ is:

$$H(\boldsymbol{x}_i,\boldsymbol{\theta}_i^*) = \frac{\Delta_i}{\lambda}\frac{\partial^2}{\partial\boldsymbol{x}^2} \ K(\boldsymbol{x},\boldsymbol{x}_i)|_{\boldsymbol{x}=\boldsymbol{x}_i}. \tag{68}$$

Since the equilibrium points are $\widetilde{\boldsymbol{x}}_j^* = \boldsymbol{x}_i$, the block diagonal components in (49) of the matrix $\boldsymbol{Q}$ are:

$$\boldsymbol{Q}_{jj} = -\mu \widetilde{p}_j H(\widetilde{\boldsymbol{x}}_j^*, \boldsymbol{\theta}_i^*) = -\mu \widetilde{p}_j H(\boldsymbol{x}_i, \boldsymbol{\theta}_i^*) = -\mu \widetilde{p}_j \frac{\Delta_i}{\lambda} \frac{\partial^2}{\partial \boldsymbol{x}^2} K(\boldsymbol{x}, \boldsymbol{x}_i)|_{\boldsymbol{x}=\boldsymbol{x}_i}. \tag{69}$$

Suppose $\Delta_i > 0$, then

$$\boldsymbol{Q}_{jj} = -\mu \widetilde{p}_j \frac{\Delta_i}{\lambda} \frac{\partial^2}{\partial \boldsymbol{x}^2} K(\boldsymbol{x}, \boldsymbol{x}_i)|_{\boldsymbol{x}=\boldsymbol{x}_i} \geq \frac{\mu \widetilde{p}_j \Delta_i}{\lambda} k_1 \boldsymbol{I}, \tag{70}$$

where $k_1$ is defined in Assumption 1. Also, the components of the matrix $\boldsymbol{R}$ in (49) are

$$\boldsymbol{R}_{jk} = \mu \widetilde{p}_j \widetilde{p}_k \left. \frac{\partial^2}{\partial \boldsymbol{x} \partial \boldsymbol{x}'} K(\boldsymbol{x}, \boldsymbol{x}') \right|_{\boldsymbol{x}=\boldsymbol{x}_j^*, \boldsymbol{x}'=\boldsymbol{x}_k^*} = \mu \widetilde{p}_j \widetilde{p}_k \boldsymbol{R}_0, \tag{71}$$

where

$$\boldsymbol{R}_0 := \left. \frac{\partial^2}{\partial \boldsymbol{x} \partial \boldsymbol{x}'} K(\boldsymbol{x}, \boldsymbol{x}') \right|_{\boldsymbol{x}=\boldsymbol{x}_i, \boldsymbol{x}'=\boldsymbol{x}_i}. \tag{72}$$

From Assumption 1, we have

$$\boldsymbol{R}_0 \geq k_3 \boldsymbol{I}. \tag{73}$$

**Case $\Delta_i > 0$.** It follows from (70) that $\boldsymbol{Q} > 0$. Also, from the definition of $\boldsymbol{R}$ in (49), $\boldsymbol{R} \geq 0$. Therefore, for any $\alpha \geq 0$, the matrix $\boldsymbol{D}(\alpha)$ in (48) is bounded below by

$$\boldsymbol{D}(\alpha) = (\alpha + \lambda)(\alpha \boldsymbol{I} + \boldsymbol{Q}) + \boldsymbol{R} \geq \lambda \boldsymbol{Q} > 0.$$

Hence, for $\psi(\alpha)$ in (48), we have $\psi(\alpha) \neq 0$. Thus, $\psi(\alpha)$ has no roots when $\alpha \geq 0$. From Lemma 5, the system is locally stable for sufficiently small $\mu_d$. This proves case (a) of Theorem 1.

**Case $\Delta_i < 0$ and $|N_i| \geq 2$.** In this case, (69) and Assumption 1 shows that

$$-\boldsymbol{Q} \in [q_1, q_2]\boldsymbol{I}, \tag{74}$$

where

$$q_1 = -\frac{\mu \Delta_i k_1}{\lambda} \min_{j \in N_i} \widetilde{p}_j, \quad q_2 = -\frac{\mu \Delta_i k_2}{\lambda} \max_{j \in N_i} \widetilde{p}_j. \tag{75}$$

Since $\Delta_i < 0$, $q_2 \geq q_1 > 0$. For $\boldsymbol{D}(\alpha)$ in (48) and $\alpha \geq 0$, let

$$\rho_{\min}(\boldsymbol{D}(\alpha)) = \min_{\|\boldsymbol{v}\|=1} \boldsymbol{v}^{\mathsf{T}} \boldsymbol{D}(\alpha) \boldsymbol{v}, \tag{76}$$

which is also the minimum eigenvalue of $\boldsymbol{D}(\alpha)$. Note that $\rho_{\min}(\boldsymbol{D}(\alpha))$ is continuous in $\alpha$.

Since $\boldsymbol{R}$ has the components (71) and $|N_i| \geq 2$, the matrix $\boldsymbol{R}$ is rank-deficient. Therefore, by selecting any vector $\boldsymbol{v}$ in the null space of $\boldsymbol{R}$, we obtain

$$\rho_{\min}(\boldsymbol{D}(\alpha)) \geq (\alpha + \lambda)(\alpha - q_2). \tag{77}$$

In particular, at $\alpha = 0$,

$$\rho_{\min}(\boldsymbol{D}(0)) = -\lambda q_2 < 0. \tag{78}$$

Also, since $\boldsymbol{R} \geq 0$,

$$\rho_{\min}(\boldsymbol{D}(\alpha)) \geq (\alpha + \lambda)(\alpha - q_1), \tag{79}$$

and

$$\rho_{\min}(\boldsymbol{D}(\alpha)) > 0, \tag{80}$$

for $\alpha > q_1$. Hence, there must be an $\alpha \geq 0$ where $\rho_{\min}(\boldsymbol{D}(\alpha)) = 0$, which implies that $\psi(\alpha) = 0$ where $\psi(\alpha)$ is the polynomial in (48). It follows that $\psi(\alpha)$ has root with $\alpha > 0$ and by Lemma 5, the equilibrium point $(\boldsymbol{\theta}_i^*, \widetilde{\boldsymbol{X}}_i^*)$ is locally unstable for all $\eta_d$ sufficiently small. This proves case (b) of Theorem 1.

**Case $\Delta_i \leq 0$ and $|N_i| = 1$.** When $|N_i| = 1$, there is a single generated point. WLOG suppose the single element in $N_i$ is $j = 1$. In this case, In this case, (69) and Assumption 1 shows that

$$- \boldsymbol{Q} \in [q_1, q_2] \boldsymbol{I}, \tag{81}$$

where

$$q_1 = -\frac{\mu \Delta_i k_1 \widetilde{p}_1}{\lambda} \quad q_2 = -\frac{\mu \Delta_i k_2 \widetilde{p}_1}{\lambda}. \tag{82}$$

Also, (71) shows that

$$\boldsymbol{R} = \boldsymbol{R}_{11} = \mu \widetilde{p}_1 \widetilde{p}_1 \boldsymbol{R}_0 \in \mu[r_1, k_2] \boldsymbol{I}. \tag{83}$$

where

$$r_1 = \mu \widetilde{p}_1^2 k_3, \qquad r_2 = \mu \widetilde{p}_1^2 k_4. \tag{84}$$

Therefore, for the matrix $\boldsymbol{D}(\alpha)$ in (48), and $\alpha \geq 0$,

$$\rho_{\min}(\boldsymbol{D}(\alpha)) \geq (\alpha + \lambda)(\alpha - q_2) + r_1 = \alpha^2 + (\lambda - q_2)\alpha + r_1 - q_2\lambda. \tag{85}$$

This polynomial will have no non-negative roots $\alpha \geq 0$, if

$$\lambda - q_2 > 0 \text{ and } r_1 - q_2\lambda > 0. \tag{86}$$

Using (82) and (84), this condition is equivalent to

$$\mu \Delta_i k_2 \widetilde{p}_1 + \min\{\lambda^2, \mu \widetilde{p}_1^2 k_4\} > 0. \tag{87}$$

In this case, $\rho_{\min}(\boldsymbol{D}(\alpha)) > 0$ for all $\alpha \geq 0$ and $\psi(\alpha)$ has no non-negative roots. From Lemma 5, the equilibrium point $(\theta_i^*, \widetilde{\boldsymbol{X}}_i^*)$ is locally stable for all $\eta_d$ sufficiently small. This proves case (c) of Theorem 1.

Similarly, taking an upper bound: the matrix $\boldsymbol{D}(\alpha)$ in (48), and $\alpha \geq 0$,

$$\rho_{\min}(\boldsymbol{D}(\alpha)) \leq (\alpha + \lambda)(\alpha - q_1) + r_2 = \alpha^2 + (\lambda - q_1)\alpha + r_2 - q_1\lambda. \tag{88}$$

This polynomial will have a positive root $\alpha$ if

$$\lambda - q_1 > 0 \text{ or } r_1 - q_1\lambda > 0. \tag{89}$$

Using (82) and (84), this condition is equivalent to

$$\mu \Delta_i k_1 \widetilde{p}_1 + \min\{\lambda^2, \mu \widetilde{p}_1^2 k_3\} < 0. \tag{90}$$

In this case, $\rho_{\min}(\boldsymbol{D}(\alpha)) = 0$ for some $\alpha > 0$. From Lemma 5, the equilibrium point $(\theta_i^*, \widetilde{\boldsymbol{X}}_i^*)$ is locally unstable for all $\eta_d$ sufficiently small. This proves case (d) of Theorem 1.

## D   Proof of Corollary 1

First, we show that $\mathbb{P}_g = \mathbb{P}_r$ is a stable local equilibrium. This situation can only occur when, for each generated point $j$

$$\widetilde{\boldsymbol{x}}_j^* = x_i \text{ and } \widetilde{p}_j = p_i, \tag{91}$$

for some $i$. Moreover, each true point $\boldsymbol{x}_i$ must have exactly one generated point $j$ with (91). Otherwise, there would be at least one true point $\boldsymbol{x}_i$ with no generated points and $\mathbb{P}_r \neq \mathbb{P}_g$. Thus, we have $|N_i| = 1$ and $\Delta_i = 0$ for all $i$. This condition satisfies (19) so the equilibria are locally stable.

Now consider any equilibrium points $\widetilde{\boldsymbol{X}} = \{\widetilde{\boldsymbol{x}}_j^*\}$ where $\text{supp}(\mathbb{P}_g) \subseteq \text{supp}(\mathbb{P}_r)$. Then, at least one true point $\boldsymbol{x}_i$ must have more than one generated point, $j$, with $\widetilde{\boldsymbol{x}}_j^* = \boldsymbol{x}_i$. That is, $|N_i| \geq 2$. Also, since the point masses are uniform, we will have

$$\Delta_i = p_i - \sum_{j \in N_i} \widetilde{p}_j = \frac{1}{N}(1 - |N_i|) < 0.$$

From Theorem 1, this equilibrium is not stable.

# E   Proof of Theorem 2

We will prove the theorem under somewhat more general assumptions on the kernel $K(\boldsymbol{x}, \boldsymbol{x}')$ as described in the following three assumptions.

**Assumption 2.** The kernel $K(\boldsymbol{x}, \boldsymbol{x}')$ satisfies $K(\boldsymbol{x}, \boldsymbol{x}') \in [0, 1]$ for all $\boldsymbol{x}, \boldsymbol{x}'$ with $K(\boldsymbol{x}, \boldsymbol{x}) = 1$ for all $\boldsymbol{x}$. In addition, $\lim_{\boldsymbol{x}'} K(\boldsymbol{x}, \boldsymbol{x}') = 0$ as $\|\boldsymbol{x}'\| \to \infty$.

The next assumption is somewhat technical, although its role will be clear in the proof.

**Assumption 3.** In an isolated region $V_i$ around the true point $\boldsymbol{x}_i$, there exists a set of distinct generated points $\widetilde{\boldsymbol{X}}_i = \{\widetilde{\boldsymbol{x}}_j, j \in N_i\}$ such that

$$\frac{1}{2} \sum_{j \neq k} \widetilde{p}_j \widetilde{p}_k K(\widetilde{\boldsymbol{x}}_j, \widetilde{\boldsymbol{x}}_k) < p_i \sum_j \widetilde{p}_j K(\boldsymbol{x}_i, \widetilde{\boldsymbol{x}}_j), \tag{92}$$

where the summations are over $j, k \in N_i$.

The final assumption requires a definition. Given a set of points $\widetilde{\boldsymbol{X}} = \{\widetilde{\boldsymbol{x}}_j, j = 1, \ldots, N\}$, let $M(\widetilde{\boldsymbol{X}})$ be the matrix with block components

$$M(\widetilde{\boldsymbol{X}})_{ij} = \frac{\partial^2}{\partial \boldsymbol{x}, \boldsymbol{x}'} K(\boldsymbol{x}, \boldsymbol{x}')|_{\boldsymbol{x}=\widetilde{\boldsymbol{x}}_i, \boldsymbol{x}=\widetilde{\boldsymbol{x}}_j}. \tag{93}$$

**Assumption 4.** For any finite set of points $\widetilde{\boldsymbol{X}}$, $M(\widetilde{\boldsymbol{X}})$ in (93) is full rank.

**Lemma 6.** *Consider the local squared MMD distance $J_i(\widetilde{\boldsymbol{X}}_i)$ in (37). Under Assumption 2 and Assumption 3, there exists at least one local minima $\widetilde{\boldsymbol{X}}_i^* = \{\widetilde{\boldsymbol{x}}_j^*, j \in N_i\}$ of $J_i(\widetilde{\boldsymbol{X}}_i)$ with $\|\widetilde{\boldsymbol{x}}_j^* - \boldsymbol{x}_i\| < \infty$ for all $j \in N_i$. In addition, the values $\widetilde{\boldsymbol{x}}_j^*$ are distinct for different $j \in N_i$.*

*Proof.* Using Assumption 2, we can rewrite the the local cost function (37) as

$$J_i(\widetilde{\boldsymbol{X}}_i) := J_0 - \sum_{j \in N_i} p_i \widetilde{p}_j K(\boldsymbol{x}_i, \widetilde{\boldsymbol{x}}_j) + \frac{1}{2} \sum_{j \neq k} \widetilde{p}_j \widetilde{p}_k K(\widetilde{\boldsymbol{x}}_j, \widetilde{\boldsymbol{x}}_k), \tag{94}$$

where

$$J_0 := \frac{1}{2} \left( p_i^2 + \sum_{j \in N_i} \widetilde{p}_j^2 \right). \tag{95}$$

By Assumption 3, there exists at least one $\widetilde{\boldsymbol{X}}_i$ such that

$$J_i(\widetilde{\boldsymbol{X}}) \leq J_0 - \epsilon, \tag{96}$$

for some $\epsilon > 0$. Now consider any limit of points $\widetilde{\boldsymbol{x}}_j \to \infty$ for all $j$. From (94), we have

$$\liminf_{\widetilde{\boldsymbol{x}}_j} J_i(\widetilde{\boldsymbol{X}}_i) \overset{(a)}{=} J_0 + \frac{1}{2} \liminf_{\widetilde{\boldsymbol{x}}} \sum_{j \neq k} \widetilde{p}_j \widetilde{p}_k K(\widetilde{\boldsymbol{x}}_j, \widetilde{\boldsymbol{x}}_k) \overset{(b)}{\geq} J_0, \tag{97}$$

where (a) follows from Assumption 2 that $K(\boldsymbol{x}_i, \widetilde{\boldsymbol{x}}_j) \to 0$ and (b) follows from the fact that $K(\widetilde{\boldsymbol{x}}_j, \widetilde{\boldsymbol{x}}_k) \geq 0$ for all $\widetilde{\boldsymbol{x}}_j$ and $\widetilde{\boldsymbol{x}}_j$. Since (97) shows that $\liminf J(\widetilde{\boldsymbol{x}}) \geq J_0$ as $\widetilde{\boldsymbol{x}}_j \to \infty$ and (96) shows that there is a point with $J_i(\widetilde{\boldsymbol{X}}_i) < J_0 - \epsilon$, there must be at least one local minimum with finite coordinates. $\square$

We now state a more general version of Theorem 2.

**Theorem 4.** *Fix a region $V_i$ and consider the dynamical system (12) with $|N_i| \geq 2$. If the kernel satisfies Assumptions 2—4. the dynamical system has at least one equilibrium with with $\widetilde{\boldsymbol{X}}_i^* = \{\widetilde{\boldsymbol{x}}_j^*, j \in N_i\}$ where*

$$\|\widetilde{\boldsymbol{x}}_j^* - \boldsymbol{x}_i\|^2 < \infty, \tag{98}$$

*for all $j \in N_i$, the $\widetilde{\boldsymbol{x}}_j^*$ are distinct for different $j \in N_i$ and the equilibrium point is locally stable for sufficiently small $\eta_d$.*

*Proof.* From Lemma 6, there exists a local minimum $\widetilde{\boldsymbol{X}}_i^* = \{\widetilde{\boldsymbol{x}}_j^*\}$ satisfying (98). From Lemma 3, there exists a $\boldsymbol{\theta}_i^*$ such that $(\boldsymbol{\theta}_i^*, \widetilde{\boldsymbol{X}}_i^*)$ is an equilibrium point of (12). So, it remains to show that the equilibrium point is locally stable. Since $\widetilde{\boldsymbol{X}}_i^*$ is a local minima of $J_i(\widetilde{\boldsymbol{X}})$ we have

$$\left. \frac{\partial^2 J_i(\widetilde{\boldsymbol{X}}_i)}{\partial \widetilde{\boldsymbol{x}}_j^2} \right|_{\widetilde{\boldsymbol{X}}_i = \widetilde{\boldsymbol{X}}_i^*} \geq \boldsymbol{0}. \tag{99}$$

Hence, from (40), we have

$$-H(\widetilde{\boldsymbol{x}}_j^*, \boldsymbol{\theta}_i^*) = \frac{1}{\lambda} \left. \frac{\partial^2 J_i(\widetilde{\boldsymbol{X}}_i)}{\partial \widetilde{\boldsymbol{x}}_j^2} \right|_{\widetilde{\boldsymbol{X}}_i = \widetilde{\boldsymbol{X}}_i^*} \geq \boldsymbol{0}. \tag{100}$$

Thus, the matrices $\boldsymbol{Q}_{jj}$ in (49) are positive semi-definite, and we have $\boldsymbol{Q} \geq \boldsymbol{0}$. Also, since the points $\widetilde{\boldsymbol{x}}_j^*$ are distinct, Assumption 4 shows that the matrix $\boldsymbol{R}$ in (49) satisfies $\boldsymbol{R} > \boldsymbol{0}$. Hence, for all $\alpha \geq 0$, the matrix $\boldsymbol{D}(\alpha)$ in (48) satisfies

$$\boldsymbol{D}(\alpha) = (\alpha + \lambda)(\alpha \boldsymbol{I} + \boldsymbol{Q}) + \boldsymbol{R} \geq \boldsymbol{R} > \boldsymbol{0}. \tag{101}$$

It follows that

$$\psi(\alpha) = \det(\boldsymbol{D}(\alpha)) \neq 0,$$

and $\psi(\alpha)$ has no real non-negative roots. The theorem now follows from Lemma 5. $\qquad\square$

We can now prove Theorem 2 as a special case of Theorem 4.

**Proof of Theorem 2**  To apply Theorem 4, we first shows the RBF kernel (7) satisfies Assumptions 2—4.

Assumption 2: This assumptions follows directly from the form of the RBF kernel (7).

Assumption 3: Given a set $\boldsymbol{U} = \{\boldsymbol{u}_1, \ldots, \boldsymbol{u}_K\} \subset \mathbb{R}^d$, with $\|\boldsymbol{u}_j\| = 1$ for all $j$, define

$$\rho_{\min}(\boldsymbol{U}) = \max_{j \neq k} \boldsymbol{u}_j^\mathsf{T} \boldsymbol{u}_k, \tag{102}$$

which is the maximum angle cosine between two unit vectors in the set. Select any $\delta < 1/2$ and set

$$N_{\max} = \max |\boldsymbol{U}| \text{ s.t. } \rho_{\min}(\boldsymbol{U}) \leq \delta, \tag{103}$$

which is the maximum cardinality of the set while keeping the angle cosine less than $\delta$. Now assume $|N_i| \leq N_{\max}$. The bound (103) states that we can find at least $|N_i|$ unit vectors $\boldsymbol{u}_j, j = 1, \ldots, |N_i|$ such that

$$\boldsymbol{u}_j^\mathsf{T} \boldsymbol{u}_k < \delta < \frac{1}{2} \tag{104}$$

for all $j \neq k$. Since $\delta < 1/2$, we find an $r$ such that

$$\frac{1}{2} e^{-r^2(1-\delta)} \sum_j \widetilde{p}_j \leq p_i e^{-r^2/2}. \tag{105}$$

Take the generated vectors as

$$\widetilde{\boldsymbol{x}}_j = \boldsymbol{x}_i + r\sigma \boldsymbol{u}_j. \tag{106}$$

Then, for the RBF kernel (7), we have

$$K(\boldsymbol{x}_i, \widetilde{\boldsymbol{x}}_j) = e^{-\|r\sigma \boldsymbol{u}_j\|^2/(2\sigma^2)} = e^{-r^2/2}. \tag{107}$$

Also, the distance between any two generated points $\widetilde{\boldsymbol{x}}_j$ and $\widetilde{\boldsymbol{x}}_k$ with $j \neq k$ is

$$\|\widetilde{\boldsymbol{x}}_j - \widetilde{\boldsymbol{x}}_k\|^2 = 2\sigma^2 r^2(1 - \boldsymbol{u}_j^\mathsf{T} \boldsymbol{u}_k) \geq 2\sigma^2 r^2(1 - \delta), \tag{108}$$

and hence

$$K(\widetilde{\boldsymbol{x}}_j, \widetilde{\boldsymbol{x}}_k) \leq e^{-r^2(1-\delta)}. \tag{109}$$

We can then verify the bound in (92):

$$\frac{1}{2}\sum_{j\neq k}\widetilde{p}_j\widetilde{p}_k K(\widetilde{\boldsymbol{x}}_j,\widetilde{\boldsymbol{x}}_k) \overset{(a)}{\leq} \frac{1}{2}\sum_{j\neq k}\widetilde{p}_j\widetilde{p}_k e^{-r^2(1-\delta)}$$

$$\overset{(b)}{\leq} \frac{1}{2}e^{-r^2(1-\delta)}\left(\sum_j \widetilde{p}_j\right)^2 \overset{(c)}{\leq} e^{-r^2/2}p_i\sum_j \widetilde{p}_j \overset{(d)}{\leq} p_i\sum_j \widetilde{p}_j K(\boldsymbol{x}_i,\widetilde{\boldsymbol{x}}_j), \quad (110)$$

where (a) follows from (109); (b) follows since we added a positive term; (c) follows from (105); and (d) follows form (107). This proves Assumption 3.

Assumption 4: Using the moment generating function of the multi-variate normal distribution, the RBF kernel (7) can be written as

$$K(\boldsymbol{x},\boldsymbol{x}') = \int a(\boldsymbol{x},\boldsymbol{\xi})^* a(\boldsymbol{x}',\boldsymbol{\xi})\phi(\boldsymbol{\xi})\,d\xi, \quad (111)$$

where

$$a(\boldsymbol{x},\boldsymbol{\xi}) = e^{i\boldsymbol{\xi}^*\boldsymbol{x}}, \quad \phi(\boldsymbol{\xi}) = \frac{\sigma^d}{(2\pi)^{d/2}}e^{-\sigma^2\|\boldsymbol{\xi}\|^2/2}. \quad (112)$$

Thus,

$$\frac{\partial^2}{\partial\boldsymbol{x}\partial\boldsymbol{x}'}K(\boldsymbol{x},\boldsymbol{x}') = \int g(\boldsymbol{x},\boldsymbol{\xi})^* g(\boldsymbol{x}',\boldsymbol{\xi})\phi(\boldsymbol{\xi})\,d\xi, \quad (113)$$

where

$$g(\boldsymbol{x},\boldsymbol{\xi}) = \boldsymbol{\xi}e^{i\boldsymbol{\xi}^*\boldsymbol{x}}. \quad (114)$$

For any distinct $\widetilde{\boldsymbol{x}}_j$, $j = 1,\ldots,N$, we have that $g(\widetilde{\boldsymbol{x}}_j,\boldsymbol{\xi})$ are linearly independent functions over $\boldsymbol{\xi}$. Hence, the matrix $M(\boldsymbol{X})$ in (93) must be full rank.

Proof of the theorem: Since the kernel satisfies Assumptions 2—4, We can thus apply Theorem 4 to find a local stable equilibrium with finite distance (98). We only have to prove that the distance scales with $\sigma$. To this end, suppose that $\widetilde{\boldsymbol{X}}_i^{(1)} = \{\widetilde{\boldsymbol{x}}_j^{(1)}\}$ is a locally minima of $J_i(\boldsymbol{X}_i)$ for the RBF kernel with $\sigma = \sigma_1$ for some $\sigma_1$. Then, given any $\sigma_2 > 0$, we can take a new set of points

$$\widetilde{\boldsymbol{x}}_j^{(2)} = \boldsymbol{x}_i + \frac{\sigma_2}{\sigma_1}(\widetilde{\boldsymbol{x}}_j^{(1)} - \boldsymbol{x}_i), \quad (115)$$

meaning that we simply scale the distances of the points $\widetilde{\boldsymbol{x}}_j$ from $\boldsymbol{x}_i$ by a factor $\sigma_2/\sigma_1$. Then, it is easily verified that $\widetilde{\boldsymbol{x}}_j^{(2)}$ will also be local minima of $J_i(\boldsymbol{X}_i)$ with the RBF kernel with width $\sigma = \sigma_2$.

## F    Proof of Theorem 3

**Lemma 7.** *For $\lambda > 0$ sufficiently small, there exists an $v_0 > 0$ such that*

$$v_0 = -\eta_g\eta_d\sum_{j=0}^{\infty}\rho^j\phi'(jv_0), \quad \rho = 1 - \eta_d\lambda. \quad (116)$$

*Proof.* Consider the function:

$$F(v,\rho) = v + \eta_g\eta_d\sum_{j=0}^{\infty}\rho^j\phi'(jv_0). \quad (117)$$

For any $\rho$,

$$\lim_{v\to\infty}F(v,\rho) = \lim_{v\to\infty}v = \infty.$$

Also, for $\rho = 1$,

$$\lim_{v\to 0}F(v,1) = \lim_{v\to 0}\eta_d\sum_{j=0}^{\infty}\phi'(jv)$$

$$= \eta_d\lim_{v\to 0}\frac{1}{v}\int_0^{\infty}\phi'(u)\,du = -\eta_d\lim_{v\to 0}\frac{1}{v}\phi(0) = -\infty. \quad (118)$$

The for $\rho$ sufficiently close to $\rho = 1$, there must exists a $v$ such that $F(v, \rho) < 0$. Since, $\lim_{v \to \infty} F(v, \rho) = \infty$ and there exists a $v$ with $F(v, \rho) < 0$, and $F(v, \rho)$ is continuous, there must exist a $v_0$ such that $F(v_0, \rho) = 0$. $\qquad\square$

Now select any unit vector $\boldsymbol{u} \in \mathbb{R}^d$ and initial condition $\widetilde{\boldsymbol{x}}_0^0$. Find $v_0 > 0$ as in Lemma 7, and define $f^k(\boldsymbol{x})$ and $\widetilde{\boldsymbol{x}}_0^k$ as:

$$f^k(\boldsymbol{x}) = -\eta_d \sum_{j=0}^{\infty} \rho^j \phi(\|\boldsymbol{x} - \widetilde{\boldsymbol{x}}_0^0 - (k - 1 - j)v_0\boldsymbol{u}\|), \tag{119a}$$

$$\widetilde{\boldsymbol{x}}^k = \widetilde{\boldsymbol{x}}_0^0 + kv_0\boldsymbol{u}. \tag{119b}$$

We show that $f^k(\boldsymbol{x})$ and $\widetilde{\boldsymbol{x}}_0^k$ are solutions to (24). The update for the discriminator is:

$$f^{k+1}(\boldsymbol{x}) = -\eta_d \sum_{j=0}^{\infty} \rho^j \phi(\|\boldsymbol{x} - \widetilde{\boldsymbol{x}}_0^0 - (k - j)v_0\boldsymbol{u}\|)$$

$$= -\eta_d \phi(\|\boldsymbol{x} - \widetilde{\boldsymbol{x}}_0^0 - kv_0\boldsymbol{u}\|) - \eta_d \sum_{j=1}^{\infty} \rho^j \phi(\|\boldsymbol{x} - \widetilde{\boldsymbol{x}}_0^0 - (k - j)v_0\boldsymbol{u}\|)$$

$$= -\eta_d \phi(\|\boldsymbol{x} - \widetilde{\boldsymbol{x}}_0^k\|) - \rho\eta_d \sum_{j=0}^{\infty} \rho^j \phi(\|\boldsymbol{x} - \widetilde{\boldsymbol{x}}_0^0 - (k - j - 1)v_0\boldsymbol{u}\|)$$

$$= -\eta_d K(\boldsymbol{x}, \widetilde{\boldsymbol{x}}_0^k) + \rho f^k(\boldsymbol{x}). \tag{120}$$

Hence, $f^k(\boldsymbol{x})$ satisfies the update (24a). Also, observe that the gradient of the discriminator in (119) is:

$$\nabla f^{k+1}(\widetilde{\boldsymbol{x}}_0^k) = -\eta_d \sum_{j=0}^{\infty} \rho^j \frac{\partial}{\partial \boldsymbol{x}} \left[ \phi(\|\boldsymbol{x} - \widetilde{\boldsymbol{x}}_0^0 - (k - j)v_0\boldsymbol{u}\|) \right]_{\boldsymbol{x} = \widetilde{\boldsymbol{x}}_0^k}$$

$$= -\eta_d \sum_{j=0}^{\infty} \rho^j \frac{\partial}{\partial \boldsymbol{x}} \left[ \phi(\|\boldsymbol{x} - \widetilde{\boldsymbol{x}}_0^{k-j}\|) \right]_{\boldsymbol{x} = \widetilde{\boldsymbol{x}}_0^k}$$

$$= -\eta_d \sum_{j=0}^{\infty} \rho^j \phi'(\|\widetilde{\boldsymbol{x}}_0^k - \widetilde{\boldsymbol{x}}_0^{k-j}\|) \frac{(\widetilde{\boldsymbol{x}}_0^k - \widetilde{\boldsymbol{x}}_0^{k-j})}{\|\widetilde{\boldsymbol{x}}_0^k - \widetilde{\boldsymbol{x}}_0^{k-j}\|}$$

$$= -\eta_d \sum_{j=0}^{\infty} \rho^j \phi'(jv_0)\boldsymbol{u} = \frac{\eta_d}{\eta_g}v_0\boldsymbol{u}, \tag{121}$$

where the last step follows from (116). Hence, for $\widetilde{\boldsymbol{x}}_0^k$ defined in (119), we have

$$\widetilde{\boldsymbol{x}}_0^{k+1} = \widetilde{\boldsymbol{x}}_0^k + \eta_g\eta_d v_0\boldsymbol{u} = \widetilde{\boldsymbol{x}}_0^k + \eta_g\nabla f^k(\widetilde{\boldsymbol{x}}_0^k). \tag{122}$$

Hence, $\widetilde{\boldsymbol{x}}_0^k$ defined in (119) satisfies the update (24b).

## G   Approximate Isolated Points

As stated in Section 2, the isolated assumption (10) may be too strict to achieve exactly in practice. In this section, we briefly consider a weaker version of this assumption. To state the approximation assumption, define

$$g^k(\boldsymbol{x}) := \frac{\partial f^k(\boldsymbol{x})}{\partial \boldsymbol{x}}, \quad G(\boldsymbol{x}, \boldsymbol{x}') = \frac{\partial K(\boldsymbol{x}, \boldsymbol{x}')}{\partial \boldsymbol{x}}. \tag{123}$$

The updates in the local region $V_i$ under the perfect isolation assumption (10) can then be written as

$$g^{k+1}(\boldsymbol{x}) = g^k(\boldsymbol{x}) + \eta_d \left( p_i G(\boldsymbol{x}, \boldsymbol{x}_i) - \sum_{j \in N_i} \widetilde{p}_j G(\boldsymbol{x}, \widetilde{\boldsymbol{x}}_j^k) - \lambda g^k(\boldsymbol{x}) \right) \tag{124a}$$

$$\widetilde{\boldsymbol{x}}_j^{k+1} = \widetilde{\boldsymbol{x}}_j^k + \eta_g g^k(\widetilde{\boldsymbol{x}}_j^k). \tag{124b}$$

Now, instead of (10), we suppose that there are neighborhoods $V_i$ around each sample $\boldsymbol{x}_i$ such that

$$\|G(\boldsymbol{x}, \boldsymbol{x}')\| \leq \epsilon \text{ for all } \boldsymbol{x} \in V_i \text{ and } \boldsymbol{x}' \in V_j \text{ for all } i \neq j, \tag{125}$$

for some $\epsilon \geq 0$. In this case, we call the set of neighborhoods $V_i$, $\epsilon$-*isolated neighborhoods*. Note that under assumption (10), the bound (125) will hold with $\epsilon = 0$. So, for $\epsilon > 0$, (125) is weaker than (10).

Next, (5) and (9) can be written as

$$g^{k+1}(\boldsymbol{x}) = g^k(\boldsymbol{x}) + \eta_d \left( \sum_{i=1}^{N_r} p_i G(\boldsymbol{x}, \boldsymbol{x}_i) - \sum_{j=1}^{N_g} \widetilde{p}_j G(\boldsymbol{x}, \widetilde{\boldsymbol{x}}_j^k) - \lambda g^k(\boldsymbol{x}) \right) \tag{126a}$$

$$\widetilde{\boldsymbol{x}}_j^{k+1} = \widetilde{\boldsymbol{x}}_j^k + \eta_g g^k(\widetilde{\boldsymbol{x}}_j^k). \tag{126b}$$

Now, fix a true point $\boldsymbol{x}_i$. We can write (126a) as

$$g^{k+1}(\boldsymbol{x}) = g^k(\boldsymbol{x}) + \eta_d \left( p_i G(\boldsymbol{x}, \boldsymbol{x}_i) - \sum_{j \in N_i} \widetilde{p}_j G(\boldsymbol{x}, \widetilde{\boldsymbol{x}}_j^k) - \lambda g^k(\boldsymbol{x}) \right) + \eta_d v^k(\boldsymbol{x}), \tag{127}$$

where $v^K(\boldsymbol{x})$ is the term from other neighborhoods:

$$v^k(\boldsymbol{x}) := \sum_{k \neq i} p_i G(\boldsymbol{x}, \boldsymbol{x}_i) - \sum_{j \notin N_i} \widetilde{p}_j G(\boldsymbol{x}, \widetilde{\boldsymbol{x}}_j^k). \tag{128}$$

From (125), for all $\boldsymbol{x} \in V_i$, the term $v^k(\boldsymbol{x})$ can be bounded as

$$\|v^k(\boldsymbol{x})\| \leq \sum_{k \neq i} p_i \epsilon + \sum_{j \notin N_i} \widetilde{p}_j \epsilon \leq 2\epsilon. \tag{129}$$

Thus, the local dynamical system in the region $V_i$ is

$$g^{k+1}(\boldsymbol{x}) = g^k(\boldsymbol{x}) + \eta_d \left( p_i G(\boldsymbol{x}, \boldsymbol{x}_i) - \sum_{j \in N_i} \widetilde{p}_j G(\boldsymbol{x}, \widetilde{\boldsymbol{x}}_j^k) - \lambda g^k(\boldsymbol{x}) \right) + \eta_d v^k(\boldsymbol{x}) \tag{130a}$$

$$\widetilde{\boldsymbol{x}}_j^{k+1} = \widetilde{\boldsymbol{x}}_j^k + \eta_g g^k(\widetilde{\boldsymbol{x}}_j^k). \tag{130b}$$

Hence, the system (130) is identical to the local dynamical system (126), except for a bounded term $\|v(\boldsymbol{x})\| \leq 2\epsilon$.

Now suppose that $\widetilde{\boldsymbol{X}}_i^* = \{\widetilde{\boldsymbol{x}}_j^*\}$ is a locally exponentially stable equilibrium point of the system (126) under perfect isolation (125). Then, such points will remain stable under the perturbations by $v^k(\boldsymbol{x})$ in (130). For example, using standard nonlinear systems results in [28], one can show that, for $\epsilon$ sufficiently small and $\|\widetilde{\boldsymbol{x}}_j^0 - \widetilde{\boldsymbol{x}}_j^*\|$ sufficiently small, there exists a $C \geq 0$ such that

$$\|\widetilde{\boldsymbol{x}}_j^k - \widetilde{\boldsymbol{x}}_j^*\| \leq C\epsilon \quad \forall j \in N_i, \quad \forall k \geq 0. \tag{131}$$

Hence, the solutions of (130) will remain close to the equilibrium point. The constant $C$ will, in general, depend on the eigenvalues of the linearization.

## H   Experimental Details

**RBF Kernel implementation**   As an approximation to the RBF feature map, we use the following approach detailed in [24]

$$K(\boldsymbol{x}, \boldsymbol{x}') \approx a^{\mathsf{T}}(\boldsymbol{x}) a(\boldsymbol{x}'), \tag{132}$$

where $a(\boldsymbol{x})$ is a basis function vector:

$$a(\boldsymbol{x}) = \sqrt{\frac{2}{R}} \begin{bmatrix} \cos(\boldsymbol{w}_1^{\mathsf{T}} \boldsymbol{x}) \\ \sin(\boldsymbol{w}_2^{\mathsf{T}} \boldsymbol{x}) \\ \vdots \\ \cos(\boldsymbol{w}_R^{\mathsf{T}} \boldsymbol{x}) \\ \sin(\boldsymbol{w}_R^{\mathsf{T}} \boldsymbol{x}) \end{bmatrix}, \quad \boldsymbol{w}_i \sim \mathcal{N}(0, \frac{1}{\sigma^2} \mathbf{I}). \tag{133}$$

When $R \to \infty$, $K(\boldsymbol{x}, \boldsymbol{x}') \to e^{-\|\boldsymbol{x} - \boldsymbol{x}'\|^2/(2\sigma^2)}$. In experiments involving the random features RBF kernel, we set $R = 1000$. We can see in figure 5 that this provides a good estimate of the true kernel.

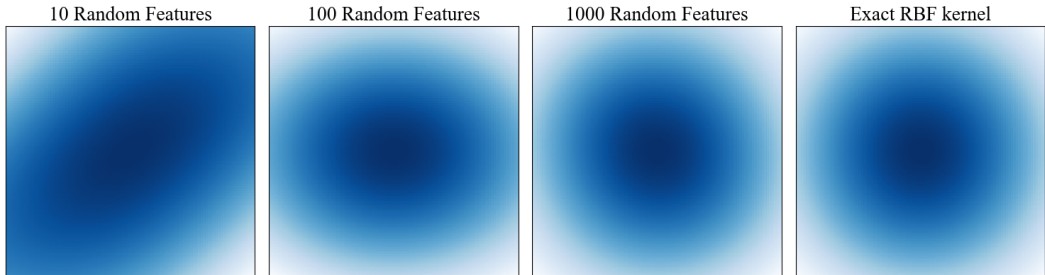

Figure 5: A heat map of different RBF kernel approximations centered at the origin. The exact RBF kernel ($\sigma = 1$) is shown on the right.

**Normalized Wasserstein Distance**  In Figure 3, the normalized Wasserstein distance is the ratio

$$\beta_k := \frac{\| \mathbb{P}_g^k - \mathbb{P}_r \|_2}{\| \mathbb{P}_g^0 - \mathbb{P}_r \|_2}, \tag{134}$$

where $\mathbb{P}_r$ is the true distribution, $\mathbb{P}_g^k$ is the generated distribution after $k$ iterations and $\| \cdot \|_2$ is the Wasserstein-2 distance. Hence $\beta_k$ in (134) is the change in the distance of the generated distance to the true distribution relative to the initial distance. In In Figure 3, we plot the normalized distance after $k = 4(10)^4$ iterations. Note that for discrete distributions, the Wasserstein-2 distance can be estimated by solving the optimal transport problem [7].

**Neural Network Discriminator**  While the focus of the paper is on kernel-based discriminators, here we rerun our two dimensional experiments with a fully connected ReLU network as our discriminator. Layer width is held constant $W = 400$, while depth is varied between $L = \{1, 2, 3, 4\}$. We note that in the wide layer (NTK) regime, the corresponding kernel width decreases as the number of layers increase [15]. We set $\lambda = 0$, $\eta_d = \eta_g = 10^{-2}$, and use 40k training steps. Just as in earlier experiments, generated points are updated directly according to equation (9).The discriminator weights are updated by standard backpropagation and gradient descent.

In Figure 6, we see that when we train a GAN with discriminator depths of one and two layers, the Wasserstein distance between distributions only changes by a small amount after training. This failure corresponds nicely to what happens in the large kernel width regime in Figure 3a. As we increase the number of layers to four, we can think of the effective kernel width of the discriminator decreasing, allowing for individual true points to be differentiated by the discriminator. We see that in this case we get much better convergence behavior. Figure 7 also supports the connection between network depth and effective kernel width. In these trajectory plots we observe large oscillations when the discriminator cannot properly distinguish between true points (catastrophic forgetting) and is far from the isolated points regime.

Lastly, in order to isolate the effect of neural network depth on convergence rate, we train a GAN with an NTK discriminator (following the closed-form solution of [5]) on a single generated point and true point. These points are initialized at a distance of 0.1 in two dimensions, and $\lambda = 0.1$, $\eta_g = \eta_r = 10^{-3}$. In figure 8, we see that by increasing the number of layers we see an increase in convergence rate.

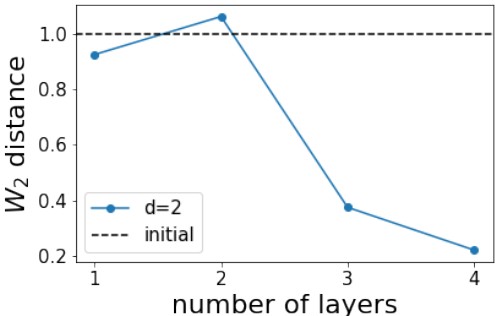

Figure 6: Median change in Wasserstein distance between true and generated distributions after 40k iterations in a two dimensional setting. Fully connected ReLU networks of different depths are used as discriminators

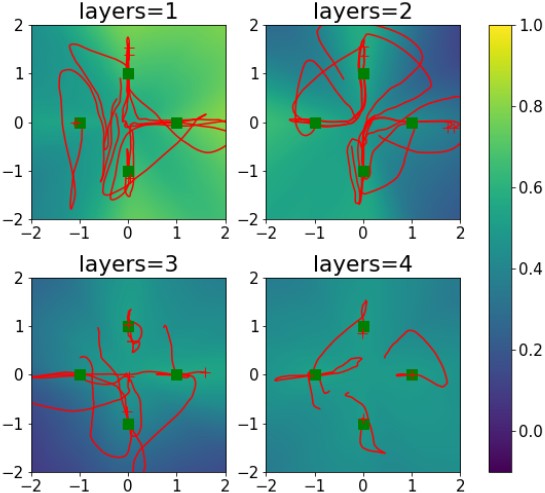

Figure 7: Behavior of joint GAN training with discriminator depth. Example trajectories of generated points over the course of training (red lines with final point marked as a cross), true distribution (green), final discriminator (blue and yellow colormap)

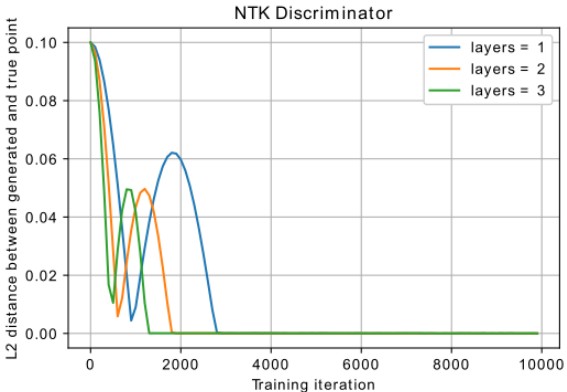

Figure 8: Convergence rates of joint GAN training under increasing NTK discriminator depth.