# OpenReview forum: "Instability and Local Minima in GAN Training with Kernel Discriminators"
_NeurIPS.cc/2022/Conference — NeurIPS 2022 Accept_

### Official Review · Reviewer_3uQN · 2022-07-08

**Rating:** 5
**Confidence:** 3
**Soundness:** 3 good
**Presentation:** 2 fair
**Contribution:** 2 fair

**Summary:**

This article propose a simple theoretical model, called the Isolated Points Model, which is analytically tractable and rigorously study the stability and convergence properties of training a GAN. The article analyze the GAN's training problems from local stability and instability, approximate mode collapse and divergence three perspectives.  Finally, this article argues a kernel based discriminator to explain the above three problems.

**Questions:**

1. Experiments are too simple to convince the theoritic prove.

2. This article does not supply a useful solution to solve the problems mentioned in the article.

3. Role of the Kernel Width in the section 6 might be a very important factor for the article, but the analysis for this factor are too simple.

4.  In the caption of Figure 1, article states that "In V4, the excess point mass the $\triangle_4$ < 0". But in the V4 of figure 1, the $\triangle_4=0.25$. Both seetings are contradictory.

5. If the RBF is set as discriminator kernel, how the RBF kernel satisfies the assumption in the equation 10.

6. Substituting equation 6 into equation 5 does not yield equation 4. Because in equation 4 is a(x_i), but the kernel is (a^T)a(x_i).

7. The article migth explain the meaning of $\delta$ in equation 1.


**Limitations:**

yes.

**Strengths And Weaknesses:**

Strengths:

1. This article proposed originality insight for the training of GAN using Isolated Points Model.

Weaknesses:
1. This article proposed originality insight for the training of GAN, but without a solution for this problems.
2. There are some problems with the misuse of symbols in the derivation section

---

> ### Author Response · Authors · 2022-08-02
> **R3 (3uQN)**
>
> Thank you for your constructive suggestions.
>
> $\textbf{Simplicity of the model:}$  This is an excellent point and we address this fact at length in the "Simplicity of the model" in the general comments above. We wanted to illustrate how failures can occur even in relatively simple multi-modal distributions. These failure modes have been observed in more complex data as well (for example see MNIST experiments in Thanh-Tung and Tran 2020), but the point of our paper is to provide a simple toy model where these failure modes can be made to occur. We believe such a simple model will help develop a more principled study and analyses of training and performance of GANs.
>
> $\textbf{Solutions and role of the kernel width:}$  As we discuss in the "Solution and future work" in the general comments above,
> the results provide interesting prescriptions on the kernel selection and the kernel width in particular. This can also provide insights into the choice of network architectures for discriminators. We leave more discussions on this for future work.
>
> $\textbf{Applicability to RBF:}$
> Regarding the applicability of the the RBF kernel to the isolated points regime, see section G of the appendix where we show that a weaker set of assumptions ($\epsilon$-isolated neighborhoods) produce similar dynamics as the original model.
>
> $\textbf{Clarification of (5):}$  Note that equation 5 comes from equation 4 when both sides are multiplied by $a(x)^T$. This transposed term comes from the definition of the discriminator in equation 2.
>
> $\textbf{$\delta$:}$  We meant the dirac-delta function, i.e., $\delta(x-x_0)$ is a point-mass at $x_0$. We will make sure to add a clearer description in the revision.

---

> > ### Comment · Reviewer_3uQN · 2022-08-06
> > **Response to Authors**
> >
> > I would like to thank the authors for their response.  I agree  the opinion of reviewer 47EW that 'Isolated Points Model' is an interesting work to analyze GANs' training. But i have the same problem as reviewer DE8a which i have mentioned in the Q2. How can we avoid instability and mode collapse in training GANs using the 'Isolated Points Model'. When we training the GANs in a real datasets like mnist or cifar10, how can we determine the width of the kernel for Discriminators?

---

> > > ### Author Response · Authors · 2022-08-08
> > > **Thank you for the response**
> > >
> > > Thank you for the response to our rebuttal, and for recognizing our work as interesting.
> > >
> > > Regarding solutions to the problems highlighted by our Isolated Points Model, we have added a new section (Section H) in the revised supplementary materials which we intend to move to the main paper if accepted. Please see the comment *Solutions via Multi-Scale Kernels* above for more details. We have added a Figure 4 which shows the proposed solution can be effective.
> > >
> > > Regarding determining the width of the kernel for discriminators, this is largely determined by the network architecture and not the dataset.
> > >
> > > We hope this addresses your concerns.

---

### Official Review · Reviewer_DE8a · 2022-07-11

**Rating:** 6
**Confidence:** 3
**Soundness:** 3 good
**Presentation:** 3 good
**Contribution:** 2 fair

**Summary:**

This paper analyzes the dynamics of training GANs under the case that the true and generated samples are discrete, finite sets, and the discriminator is kernel-based. Specifically, the paper assumes that the true samples are at least one kernel width away from each other, so called Isolated Points Model. This paper analyzes the necessary and sufficient conditions for local stability and instability, points out the existence of approximate mode collapse and divergence, and provides insights into the role of the kernel width of the kernel discriminator. Numerical experiments are also conducted to show the effect of kernel width on GAN training.

**Questions:**

- What new principles does this study bring to GAN design? In other words, how do we avoid instability and mode collapse in training GANs?

typos:
- In line 104, "generated samples" -> "true samples"

**Ethics Review Area:**

["I don’t know"]

**Limitations:**

The authors should discuss the limitations and potential negative societal impact of their work.

**Strengths And Weaknesses:**

Strengths
- This paper is well organized and well written.
- This paper is theoretically grounded. It provides a theoretical understanding of the dynamics of GAN training.


Weaknesses
- Assumptions are strong. For example, in line 110, supposing that N_i^k is constant over time is unreasonable. It is not clear to me at this time how this assumption affects the theoretical results. I would like to see the authors analyze the results when this assumption does not hold.
- Experiments are weak. Only numerical experiments with simple experimental settings are provided in this work.

---

> ### Author Response · Authors · 2022-08-02
> **R2 (DE8a):**
>
> Thank you for your suggestions regarding assumptions and experiments.
>
> $\textbf{Strong assumptions:}$ While it is true that in practice generated points may change neighborhoods, we are focused on the training behavior in its limit. If the kernel width is small enough, and the generated points converge, then eventually all generated points will either be contained within a local neighborhood or outside all local neighborhoods. Note that neighborhoods do not have to be perfectly isolated for these dynamics to hold: in supplementary section G we give a weaker $\epsilon$-isolated neighborhood definition.

---

> > ### Comment · Reviewer_DE8a · 2022-08-06
> > **Response to Authors**
> >
> > I would like to thank the authors for their response. I remain concerned about the strong assumption and the lack of potential solutions. In my opinion, if the generated points have converged, there is no need to analyze their training stability. I noticed that reviewer 3uQN have the same question as I, that is, how do we avoid instability and mode collapse in training GANs, especially on real-world datasets? Answering this question with a validated solution would greatly strengthen the contribution of this paper.

---

> > > ### Author Response · Authors · 2022-08-08
> > > **Thank you for the response**
> > >
> > > We thank you for the remarks and the questions raised.
> > >
> > > Based on your recommendation, we have now added a section (Section H) in the revised supplementary materials which we intend to move to the main paper if accepted (we are allowed an extra page in final version).
> > > Our solution is simple and surprisingly effective and follows naturally from the equilibrium analysis in the paper. Please see details in the comment *Solutions via Multi-Scale Kernels* above. Figure 4 in the supplementary materials shows that this solution can be effective.
> > >
> > > We hope this addition address your concerns.

---

### Official Review · Reviewer_47EW · 2022-07-11

**Rating:** 7
**Confidence:** 3
**Soundness:** 3 good
**Presentation:** 3 good
**Contribution:** 2 fair

**Summary:**

This paper studies the gradient descent-ascent optimization problem of GANs in the following setting:
 1. without generator, the GAN manipulating a finite set of elements;
 2. under an MMD GAN model with a fixed kernel, the kernel discriminator optimizing the corresponding IPM;
 3. in a novel framework called the Isolated Points Model where the kernel value between true data points approaches zero.

The latter assumptions enable the derivation of new convergence results. The paper particularly characterizes the existence, position and nature of equilibrium points in the gradient descent-ascent optimization around the true samples tanks to the introduction of a probability mass difference criterion. This notably allows the authors to rule out the possibility of strict mode collapse on the one hand and to identify the existence of bad local minima leading to an approximate mode collapse phenomenon. The theoretical results are then assessed experimentally on a toy dataset.

**Questions:**

 - Could the authors further evaluate and discuss the realism of the introduced Isolated Point Model? In particular, how might it relate to the theory of NTKs as mentioned in the conclusion?
 - Would it be possible to highlight how the obtained results and/or the used proof techniques might serve as a basis for future work?
 - Do the authors see any relaxation of the Isolated Point Model and Assumption 1 beyond considering exponentially vanishing kernels like the RBF? For example, by constraining only the gradients of the kernel instead of its values, since the movement of the generated points only depend on its gradients?

**Limitations:**

The authors adequately present the assumptions and limitations of their study. While many of them are standard and reasonable, further discussion of the realism of the main and novel assumptions would be beneficial for the paper as it could help the reader gauge the impact of the obtained results (cf. the above sections).

**Strengths And Weaknesses:**

### Strengths

The study of the optimization procedure of GANs is a highly challenging task, as understanding the outcome of its non-convex non-concave behavior remains an open question in the literature. In this regard, **the tackled problem is relevant** to the NeurIPS community.

This **well-written paper** presents a clear proposition to advance our understanding of this difficult problem. It properly defines its analysis framework composed of assumptions (1, 2, 3) as described in the above summary. While **conditions (1, 2) are reasonable, well-motivated choices** to make the optimization problem amenable to theoretical analysis, the Isolated Point Model (3) necessarily limits the applicability of the presented results. Nonetheless, such a strong assumption is understandable given the challenge of the tackled issues and **pushes back the limits of previous convergence analyses** whose assumptions were even more restrictive.

To my knowledge, the obtained theoretical results are **clearly presented**, **novel** and **may contribute to a better understanding of GAN optimization**. The identification of precise criteria for the existence of stable and unstable equilibrium points as well as the discovery of an approximate mode collapse is very welcome and participate in deepening our knowledge of local convergence effects in GANs.

### Weaknesses

My main interrogation on the paper deals with **the significance of the presented results**. Given the restrictions imposed by the Isolated Point Model, their impact on the domain remains unclear and difficult to gauge. Indeed, the proposed Isolated Point Model is **unrealistic** in the context of GANs. While kernel discriminators may be considered via the theory of NTKs [8], these kernels are non-standard and typically do not resemble the RBF-like kernels considered in this paper. Even in the case of the RBF, **a small kernel width is not a typical choice**, as explained by the empirical results of Section 7. This lack of grounding in GAN practice is highlighted by the divergence result of Section 5, which to my knowledge do not correspond to a standard GAN behavior. I am also unable to judge whether the introduced framework might also serve as a basis for future, less limited analyses, as this is a difficult task.

### Overall Sentiment

Given the difficulty of the tackled problem, the state of the literature on this matter, and the interesting phenomena unveiled by the authors, I think that this paper is worth publishing at NeurIPS.

Even though the chosen setting might be seen as unrealistic, I believe that the presented advances have the potential to be useful to the community, hence my positive recommendation. I would be glad to discuss this matter with the authors and the other reviewers during the discussion phase.

### Typos

 - Figure 1: I think the description of $V_4$ in the caption actually corresponds to $V_2$ in the figure.
 - Line 172: "excactly" should be "exactly".
 - Line 268: "singl" should be "single".
 - Lines 189, 214, 265 and 295: please fix the quote typesetting.

---

> ### Author Response · Authors · 2022-08-02
> **R1 (47EW):**
>
> Thank you for the overall positive comments and appreciating the novelty of the perspective.
>
> $\textbf{Realism:}$ Regarding the realism of the model, please see the comments on "Simplicity of the model".
>
> $\textbf{Applicability to practical setting:} $
> It is true that for shallow neural networks, the kernel width for the NTK is typically large.  However, it is well-known (Bietti and Mairal 2019, Huang et al. 2020) that as the number of layers increase, the kernel width may be small.  While we plan to expand upon rate of convergence in later work (see the general comment on "Solutions and Future Work"), for now we have included an example of the convergence behavior of GANs trained with an NTK-based kernel discriminator of varying depths in Figure 8 in Section I of the appendix. One can see that tuning the number of layers in a discriminator (or even dynamically adding layers) can be an important method for improving convergence. Hence, the results may have practical impact for very deep networks.
>
> $\textbf{Relaxing assumptions:}$  We discuss a potential relaxation of Assumption 1 in Appendix G. As you suggest, we only need to constrain the gradients of the kernel.  The kernel does not need to decay at any particular rate (e.g., exponential).

---

> > ### Comment · Reviewer_47EW · 2022-08-05
> > **Response to Authors**
> >
> > I would like to thank the authors for their response. Overall, I still think that this paper can be published at NeurIPS, given the difficulty of the tackled task. I agree with the authors that simplifying hypotheses are needed to analyze GANs' training dynamics and the presented method does constitute an interesting attempt at deriving new results from less unrealistic assumptions. The interpretation of the Isolated Point model as an abstraction for multi-modal distributions is especially interesting and the paper would benefit from further discussion on this matter.
> >
> > Regarding the NTK kernel width, increasing depth is indeed an important factor to take into account. However, to my knowledge, the depth growing unbounded can also produce degenerate kernels which might not be useful for generative modeling, but this is beyond the scope of this work. Nonetheless, this discussion is interesting and could be included in the paper to strengthen the discussion on the realism of the proposed framework.

---

> > > ### Author Response · Authors · 2022-08-08
> > > **Thank you for championing our paper**
> > >
> > > Thank you for recognizing the difficulty of the problem we are trying to tackle. We would really appreciate it if you could share your perspective on the hardness of this problem with the other reviewers. Although applying our abstraction to real-world datasets is desirable, it is currently beyond the scope of the current paper.
> > >
> > > We have also added a new section about solutions to mitigate the issues raised by our model. Please see our comment on *Solutions via Multi-Scale Kernels* for more details.
> > >
> > > Regarding the discussion on NTK kernel width, thank you for pointing out this degeneracy. We shall include a note on this matter in the final revision of the paper.

---

### Author Response · Authors · 2022-08-02
**General comments**

We would like to thank the reviewers for their careful reading of the paper and the insightful comments and questions.  We are glad that all the reviewers found the framework novel and appreciated the need for results on joint dynamics of training in GANs.    We address two common concerns raised by the reviewers and then address the specific concerns in individual comments. We thank both reviewers $\textbf{47EW}$ and $\textbf{3uQN}$ for pointing out that $V_4$ should be changed to $V_2$ in the caption of figure 1.

### Simplicity of the model
All the reviewers remark that the isolated points model is somewhat simplistic.  First, it should be pointed out that the model is significantly more complex than most prior results of this nature -- a point made by reviewer 47EW. For example, the Dirac-GAN (Mescheder et al. 2018) considers only a single point mass for the generator and true distribution and a linear discriminator. Secondly, we argue that the simplicity is in fact, a $\textit{strength}$.
The isolated points model provides a simple abstraction for multi-modal distributions.
In this sense, the paper illustrates that failure can occur whenever the true distribution is multi-modal and the modes are separated by distances significantly greater than the kernel width.  As such, we believe this model is the first to capture the key relation between the multi-modal nature of the distributions, the kernel width and certain failure modes.
Most importantly, few other papers have a rigorous framework that explains this particular failure mode and our goal is to provide the simplest and most clear settings where these can occur.  Finally, see Supplementary Section G on how the model can be relaxed.

### Solutions and future work
The results suggest a natural prescription for the kernels in the discriminator to avoid the undesirable behaviors. Unfortunately, due to space considerations, we were unable to analyze this potential solution, one natural method is is to develop kernels that are sharp near the origin but also heavy-tailed. Interestingly, this 'kernel shaping' has been shown to be implicitly performed by wide skip-connection networks (Huang et al. 2020). Another possibility is to dynamically increase the number of layers in a NN discriminator during training. In section I Figure 8 we show that increasing the effective depth of the NTK discriminator indeed increases the rate of local convergence. An interesting avenue of future work is to analyze these potential architectures and their rates of convergence.

---

### Author Response · Authors · 2022-08-08
**Solutions via Multi-Scale Kernels**

We appreciate that the reviewers asked about
what solutions the paper suggests.
Due to space considerations, we had originally left
the topic of solutions for future work.
But, we have now added a new section (Section H of the supplementary materials)
in the revision on "Potential Solutions via Multi-Scale Kernels". We intend to move this section to the main paper if accepted (as the page limit is increased then).  The idea is to create
kernels of the form $K(x,x') = K_1(x-x') + K_2(x-x')$ where $K_1$ has small width
and $K_2$ has a wide width. The wide
width component avoids the isolation between
points and may alleviate the approximate
mode collapse and divergence issues shown
by our results.  At the same time,
the narrow width kernel can provide fast convergence
near the true points and better discrimination
of true points that are close.  We provide some preliminary
simulations that show the effectiveness of such a solution.

Interestingly, this 'kernel shaping' has been shown to be implicitly performed by wide skip-connection networks (Huang et al. 2020).   However,
we leave it to future work to find the best
architectures to provide such multi-scale
behavior on more complex data.

---

### Meta-Review · Area_Chair_eaVF · 2022-09-02

**Recommendation:** Accept
**Confidence:** Less certain

**Metareview:**

All reviewers agree this paper presents interesting analysis and results on GAN training dynamics. However they also note several limitations of the proposed setting, namely the assumptions of kernel width and the isolated points model, restrict directly applying results to practical settings. Authors in the response have done a good job of explaining how one can use the observations from the analysis to improve stability of real world training of GANs. Overall I think this work has some promising directions towards improving our understanding of training GANs, hence I suggest acceptance.

**Award:**

No

---

### Decision · Program_Chairs · 2022-09-14

Accept